# *Coxiella burnetii* manipulates the lysosomal protease cathepsin B to facilitate intracellular success

Lauren E. Bird [1,2], Bangyan Xu [3], Andrew D. Hobbs [2],
Alexander R. Ziegler [3], Nichollas E. Scott [1], Patrice Newton [1,2],
David R. Thomas [2], Laura E. Edgington-Mitchell [3,4] ✉ &
Hayley J. Newton [1,2,4] ✉

The obligate intracellular bacterium *Coxiella burnetii* establishes an intracellular replicative niche termed the *Coxiella*-containing vacuole (CCV), which has been characterised as a bacterially modified phagolysosome. How *C. burnetii* withstands the acidic and degradative properties of this compartment is not well understood. We demonstrate that the key lysosomal protease cathepsin B is actively and selectively removed from *C. burnetii*-infected cells through a mechanism involving the Dot/Icm type IV-B secretion system effector CvpB. Overexpression of cathepsin B leads to defects in CCV biogenesis and bacterial replication, indicating that removal of this protein represents a strategy to reduce the hostility of the intracellular niche. In addition, we show that *C. burnetii* infection of mammalian cells induces the secretion of a wider cohort of lysosomal proteins, including cathepsin B, to the extracellular milieu via a mechanism dependent on retrograde traffic. This study reveals that *C. burnetii* is actively modulating the hydrolase cohort of its replicative niche to promote intracellular success and demonstrates that infection incites the secretory pathway to maintain lysosomal homoeostasis.

The mammalian lysosome is central to cellular homoeostasis, acting as a control centre for nutrient recycling and immune surveillance. Lysosomal degradation is primarily driven by >50 different hydrolases present in the lumen, and dysfunction of these enzymes is associated with a range of devastating disease pathologies including cancer, neurodegeneration and cardiovascular disease[1,2]. One major class of lysosomal proteases is the cathepsin family, which broadly serves to regulate protein degradation and turnover, thus maintaining cellular homoeostasis[3]. To regulate proteolysis, cathepsins are first synthesised in an immature zymogen form, containing an autoinhibitory pro-domain at their N- or C-terminus[4]. In the Golgi apparatus, immature cathepsins are tagged with mannose-6-phosphate, which directs their trafficking to the lysosome. Once inside this compartment, the low pH and activity of other proteases leads to cleavage of the pro-domain resulting in a mature, active cathepsin protein which participates in cellular catabolism. For obligate intracellular pathogens, avoiding lysosomal degradation is critical to a successful infection (reviewed in ref. 5). Following internalisation into a host cell, pathogens can interfere with maturation of the pathogen-containing phagosome or induce endomembrane damage to escape it, avoiding exposure to lysosomal content. The only known bacterial pathogen to successfully replicate inside a lysosome-derived compartment is the Gram-negative *Coxiella*

---

[1]Department of Microbiology and Immunology at the Peter Doherty Institute for Infection and Immunity, The University of Melbourne, Melbourne, VIC, Australia. [2]Infection Program, Monash Biomedicine Discovery Institute, Department of Microbiology, Monash University, Clayton, VIC, Australia. [3]Department of Biochemistry and Pharmacology at the Bio21 Molecular Science and Biotechnology Institute, The University of Melbourne, Parkville, VIC, Australia. [4]These authors contributed equally: Laura E. Edgington-Mitchell, Hayley J. Newton. ✉e-mail: laura.edgingtonmitchell@unimelb.edu.au; hayley.newton@monash.edu

*burnetii*. Exactly how *C. burnetii* withstands such a hostile intracellular environment is not fully understood.

*C. burnetii* is the causative agent of the global zoonosis Q fever, which can manifest in chronic and debilitating presentations such as endocarditis and chronic fatigue[6]. The primary disease reservoirs are ruminants such as cattle, sheep, and goats, and once shed to the environment, humans can become infected with *C. burnetii* through inhalation of aerosols containing even one bacterium[7]. The low infectious dose, aerosol transmission and high environmental resistance properties have led to *C. burnetii* being classified as a Category B potential bioterrorism agent by the US Centers for Disease Control and Prevention, and a Select Agent by the Federal Select Agent programme. Concerningly, analysis of *C. burnetii* seroprevalence in Australia indicates that up to 1 in 20 people have been exposed to this pathogen[8].

Once internalised, *C. burnetii* establishes a spacious and fusogenic compartment termed the *Coxiella*-containing vacuole (CCV), which is decorated with both lysosomal and autophagosomal proteins[9]. From within this replicative niche, *C. burnetii* employs a type IV-B secretion system (T4SS) to translocate bacterial effector proteins across the bacterial and CCV membranes into the host cytosol. Disruption of the T4SS abolishes *C. burnetii* intracellular, but not axenic, replication, demonstrating the importance of this system for intracellular survival[10,11]. Early studies on the *C. burnetii* intracellular lifecycle report that the CCV contains an acidic pH and degradative enzymes such as acid phosphatase, leading to the theory that *C. burnetii* resides in a bacterially-modified phagolysosome[12–15]. Increasing lysosomal pH with chloroquine abolished *C. burnetii* replication, further supporting this notion[16]. However, the extent to which the CCV diverges from an endogenous phagolysosome is currently unclear, with reports indicating that the CCV pH (~5.2) is higher than native lysosomes (~4.8) and restoring CCV pH to 4.8 is toxic to *C. burnetii*. Further, induction of lysosome biogenesis was also found to be detrimental to intracellular replication[17,18]. However, it was also demonstrated that lysosomal hydrolase activity generates essential amino acids sensed by the *C. burnetii* regulatory two-component system PmrAB, promoting initial activation of the T4SS[19]. Interestingly, perturbation of the mannose-6-phosphate trafficking pathway, which directs hydrolases to lysosomes, did not affect *C. burnetii* viability or replication but did impact the morphology of CCVs[20].

Given these reports, it is currently accepted that *C. burnetii* replicates in a proteolytically active lysosome-derived vacuole. However, the suite of lysosomal proteases present and active in the CCV has never been thoroughly examined. Here, we sought to investigate the abundance and activity of host lysosomal proteases of the cathepsin family during *C. burnetii* infection and to assess whether modulation of these proteases might impact bacterial success. Specifically, we hypothesised that *C. burnetii* might downregulate specific lysosomal proteases to create an environment that is permissive of bacterial replication. Using orthogonal experimental approaches, we demonstrate that the host lysosomal protease cathepsin B is removed from *C. burnetii*-infected cells in a T4SS-dependent manner and that overexpression of active cathepsin B is detrimental to *C. burnetii*. Furthermore, we show that this loss is dependent on a vacuolar environment established by the *C. burnetii* effector CvpB. Interestingly, we observed secretion of a suite of lysosomal proteins, including cathepsin B, into the extracellular milieu during infection via a mechanism that involves the default secretory pathway. Taken together, these data demonstrate that lysosomal proteolysis and protein trafficking are subverted during *C. burnetii* infection, significantly advancing our understanding of how this unique pathogen manipulates human lysosomal biology to cause disease.

## Results

### The lysosomal protease cathepsin B is actively removed from human cells during *Coxiella burnetii* infection

To globally assess host protease abundance during *C. burnetii* infection, we performed label-free mass spectrometry on lysates from uninfected and *C. burnetii*-infected THP-1 cells (Supplementary Fig 1A, B and Supplementary Data 1). Consistent with previous research[21], we observed infection-dependent enrichment of proteins associated with lysosomes and vacuoles (Fig. 1A and Supplementary Table 1). Supporting this, examination of proteome Z-scores indicated an increase in the abundance of cathepsin proteases during infection, with the exceptions of cathepsin C (CTSC) which showed a small reduction, and cathepsin B (CTSB), which was dramatically decreased (Fig. 1B and Supplementary Fig 1C, D). This finding was supported by immunoblotting with an antibody to the heavy chain subunit of cathepsin B over 3 days of infection, which revealed that cathepsin B was undetectable by day 2 (Fig. 1C). This corresponded with an accumulation of immature (pro-) cathepsin D (pCTSD, Fig. 1C and Supplementary Fig 1E), aligning with reports that cathepsin B is involved in the proteolytic activation of cathepsin D[22]. Via immunoblotting, we also observed reduced abundance of cathepsin C at 3 days post-infection. However, unlike cathepsin B, this protein was not completely absent (CTSC, Fig. 1C) and the decrease in abundance was not significant as assessed by quantitative proteomics analysis (Supplementary Fig 1A and Supplementary Fig 1D), leading us to focus our investigation on cathepsin B. Analysis of transcript abundance using real-time quantitative PCR (RT-qPCR) confirmed that the changes in cathepsin expression during infection were not due to reduced transcription of these proteins (Fig. 1D). To increase coverage of cathepsin B peptides identified in our mass spectrometry experiments, we performed data-independent acquisition (DIA) mass spectrometry on uninfected and infected cell lysates[23]. Using this approach, we observed that the peptide [72]VMFTEDLK[79] from the pro-domain of cathepsin B increased in abundance during infection, in contrast to decreased abundance of peptides within the mature protein (Fig. 1E). In combination with the RT-qPCR data, this suggests that mature cathepsin B protein is being lost downstream of synthesis.

Having observed differential modulation of cathepsins during *C. burnetii* infection, we utilised a pan-reactive activity-based probe to measure cysteine cathepsin activity in live cells. BMV109 covalently binds to the active site of cathepsin X, B, S, and L, and emits fluorescence upon interaction with the respective target proteases, allowing visualisation of protease activity by in-gel fluorescence[24,25]. Live cell labelling of THP-1 cells with BMV109 confirmed that cathepsin B activity is lost during infection, with other cysteine cathepsins showing increased or unchanged activity (Fig. 1F). This was supported with immunoblots using antibodies against cathepsin L, X and S, confirming that these proteases show increased or unchanged expression during infection (Supplementary Fig 1F). At the single-cell level, immunofluorescence microscopy (IF) confirmed that cathepsin B was removed from wild-type-infected cells but not neighbouring uninfected cells (Fig. 1G, H).

*C. burnetii* infection is known to alter the endosomal compartment through occupation and modulation of lysosomes[18,26]. To determine whether the decreased cathepsin B levels were due to an altered endosomal compartment, we generated THP-1 cells stably overexpressing a fluorescently-tagged transcription factor EB (TFEB-GFP). As expected, TFEB overexpression led to increased expression of cathepsin B in uninfected cells (Supplementary Fig 1G, H), consistent with an increase in lysosomal biogenesis. However, infection of these cells with *C. burnetii* again led to cathepsin B removal, suggesting that decreased levels of proteolytically active cathepsin B during infection are not a consequence of the altered endosomal compartment.

We next aimed to elucidate whether cathepsin B removal was the result of *C. burnetii* T4SS activity. To achieve this, we used strains carrying transposon insertions in the *icmL* or *icmS* genes. The *icmL*::Tn mutant harbours a non-functional T4SS and is subsequently unable to translocate bacterial effector proteins to the host cytosol, leading to a complete abolishment of intracellular replication[11]. Similarly, the *C. burnetii* chaperone pair IcmSW has been shown to promote the

 

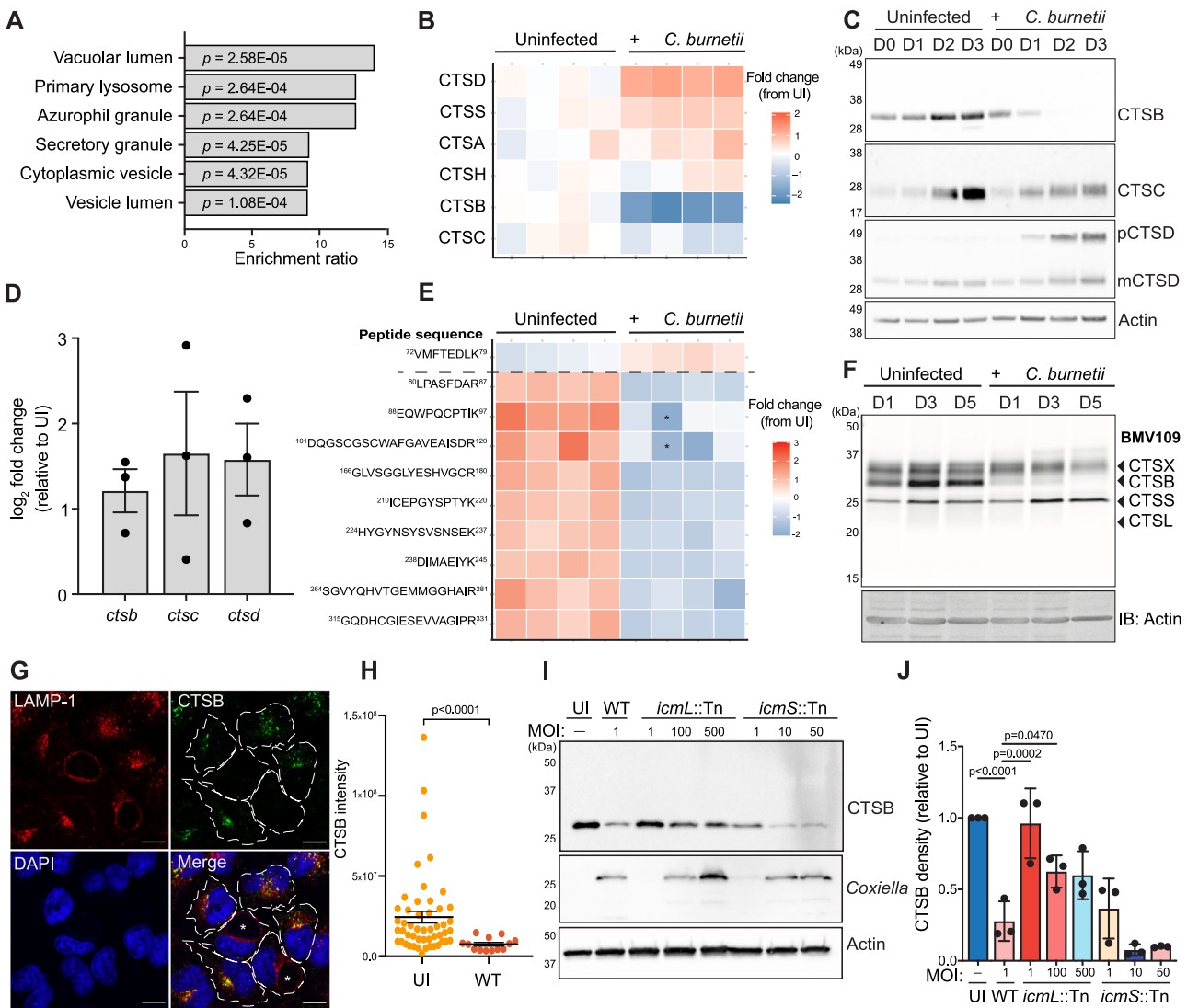

**Fig. 1 | Altered abundance and activity of lysosomal proteases during *C. burnetii* infection. A, B** THP-1 cells were uninfected (*n* = 4) or infected (*n* = 4) with *C. burnetii* (MOI 50) for 72 h prior to sample collection for mass spectrometry. **A** Gene ontology (GO) on significantly enriched proteins. GO analysis was performed using WebGestalt overrepresentation test and the cellular component functional data-base. *P*-values calculated using Fisher's exact test with BH correction for multiple comparisons. **B** Heat map of LFQ intensities of protein abundance for all cathepsins identified. Each column represents one biological replicate. **C** Immunoblot showing cathepsin expression over 3 days in uninfected or *C. burnetii*-infected THP-1 lysates. **D** qRT-PCR analysis of transcript abundance for cathepsin B, C, and D in uninfected or infected THP-1 cells after 72 h of infection. Raw values were normalised against 18S rRNA transcript abundance and are presented as normalised log₂ fold change from UI. Data reflect 3 biological replicates, error bars represent SEM. **E** Heatmap of cathepsin B peptides observed from DIA proteomic analysis of 72 h THP-1 infected cells. Peptides are ordered from N-terminus (top) to C-terminus (bottom), with horizontal dashed line indicating cleavage site at end of pro-domain. Imputed values denoted with an asterisk. **F** BMV109 labelling of active cysteine proteases at D1, 3, or 5 post infection. Gel is representative of 3 independent experiments. **G** Representative image of HeLa cells infected with *C. burnetii* (MOI 1) for 72 h and stained with antibodies against LAMP-1 and cathepsin B. Infected cells were iden-tified by LAMP-1 staining and are denoted with an asterisk. **H** Quantification of imaging in **G**. Data points represent individual cells measured within one experi-ment (UI, *n* = 50, WT, *n* = 13). Error bars represent mean + SEM (**I**, **J**) Immunoblot of THP-1 cells infected with wild-type *C. burnetii* (WT), *icmL*::Tn or *icmS*::Tn mutants at the indicated MOI. Blot is representative of three independent experiments (quantified in **J**). Error bars reflect SD. CTSB cathepsin B, CTSC cathepsin C, pCTSD pro-cathepsin D, mCTSD mature cathepsin D; Scale bar = 10 μm. Source data are provided as a Source Data file. Mass spectrometry source data are available via ProteomeXchange with identifier PXD052888.

translocation of a specific cohort of T4SS effectors, but disruption of IcmS also causes a partial replication defect[27]. We attempted to mea-sure cathepsin B signal in cells infected with *icmL*::Tn or *icmS*::Tn via IF, but had limited success due to low bacterial loads and difficulty identifying infected cells (Supplementary Fig 2A, B). We compensated for this by infecting at multiplicities of infection (MOI) up to 500X greater than wild-type and examining cathepsin B abundance at the population level with immunoblotting. Infection with *icmL*::Tn at any MOI tested led to greater retention of cathepsin B than wild-type infection (Fig. 1I, J), indicating that T4SS activity is required for

cathepsin B removal. Moreover, cathepsin B was lost during *icmS*::Tn infection at increasing MOI, signifying that translocation of the responsible effector(s) is IcmS-independent. Taken together, these data indicate that cathepsin B is removed during *C. burnetii* infection as a result of T4SS effector activity.

## Overexpression of cathepsin B is detrimental to *C. burnetii* replication and CCV biogenesis

We hypothesised that *C. burnetii* may be actively removing cathepsin B to promote formation of a replicative niche that is less hostile than a

classical host lysosome. To directly investigate the impact of cathepsin B expression on *C. burnetii*, we generated HeLa cell lines in which GFP-tagged wild-type cathepsin B (CTSB-GFP) or a catalytically inactive mutant (CTSB^C108A-GFP) were overexpressed (Fig. 2A). Labelling of these cell lines with BMV109 confirmed that cathepsin B over-expression led to increased cathepsin B activity in CTSB-GFP cells, but not CTSB^C108A-GFP cells (Fig. 2B). Immunoblotting of the same lysates demonstrated equivalence in cathepsin B expression in both over-expressing cell lines (Fig. 2B). Furthermore, infection of CTSB-overexpressing cell lines with *C. burnetii* led to a reduction, though not a complete absence, in cathepsin B (Fig. 2C). Interestingly, we observed that GFP was cleaved from the fusion protein in an infection-dependent manner (Fig. 2C, middle panel), however the relative abundance of the CTSB-GFP fusion protein (75 kDa) was unchanged during infection compared to uninfected cells.

Quantification of *C. burnetii* intracellular replication over 5 days of infection revealed that replication was lower in CTSB-GFP cells than cells expressing GFP alone at day 3 (mean difference $122.9 \pm 28.96$, $p = 0.006$) and day 5 ($457.1 \pm 123.5$, $p = 0.03$) post infection (Fig. 2D). Mutation of the cathepsin B active site (CTSB^C108A-GFP) resulted in an intermediate phenotype but did not completely rescue the growth defect.

We also examined morphology of the CCVs in each cell line. IF revealed that CCVs were smaller in CTSB-GFP cells than HeLa parent cells or cells expressing GFP alone (mean difference $52.8 \pm 10.87$, $p = 0.006$ or $45.8 \pm 10.87$, $p = 0.001$, respectively, Fig. 2E, F). Cells expressing catalytically inactive cathepsin B (CTSB^C108A-GFP) displayed an intermediate phenotype with respect to CCV area. Collectively, these data demonstrate that overexpression of cathepsin B is detrimental to *C. burnetii* replication and CCV biogenesis, indicating that active removal of this protein is an important element of *C. burnetii* intracellular success.

## Cathepsin B is retained during infection with *C. burnetii* lacking the T4SS effector CvpB

In light of our findings that cathepsin B loss is dependent on a functional Dot/Icm T4SS, we aimed to determine which *C. burnetii* effector protein(s) might be responsible for loss of cathepsin B during infection. To achieve this, we infected HeLa cells with a library of *C. burnetii* mutants with transposons or deletions in 38 known T4SS effector-encoding genes (Supplementary Data 2). Given that this library is incomplete, we also ectopically expressed individual effector proteins (fused to an N-terminal 3xFLAG tag) in HeLa cells and used IF or western blotting (WB) to visualise cathepsin B (Supplementary Data 2). These approaches in combination allowed us to assess the impact of 134 unique *C. burnetii* proteins on cathepsin B abundance. Interestingly, we observed partial retention of cathepsin B during infection with a strain lacking the effector CvpB (also referred to as Cig2, encoded by gene *CBU_0021*) (Fig. 3A, B). This phenotype was able to be complemented, with cathepsin B lost after CvpB reintroduction (*cvpB*::Tn (pFLAG-*cvpB*)) (Fig. 3A, B). CvpB (*Coxiella vacuolar protein B*) is a highly conserved *C. burnetii* effector that localises to the CCV membrane during infection[28,29] and has been partially characterised as having a key role in CCV biogenesis[9,28,30] via modulation of phosphoinositide metabolism[29]. Furthermore, as with cathepsin B loss, translocation of this effector protein through the T4SS is independent of IcmS[27]. Disruption of CvpB leads to a unique multi-vacuole phenotype whereby small CCVs, which are still capable of bacterial replication, are unable to undergo homotypic fusion[9]. Analysis of *cvpB*::Tn-infected cells with IF revealed that cathepsin B puncta were observable in a portion of cells (Fig. 3C, white arrowheads, Fig. 3D). Quantitation of cathepsin B signal intensity revealed that infection with *cvpB*::Tn did not lead to a significant reduction in cathepsin B compared to uninfected cells, contrary to the phenotype observed during infection with wild-type or the complement (Fig. 3D). Proteomic analysis further

confirmed increased abundance of cathepsin B during infection with *cvpB*::Tn compared to wild type (Fig. 3E and Supplementary Data 1). Taken together, these orthogonal experimental approaches indicate that cathepsin B is retained in cells infected with *C. burnetii* lacking the T4SS effector CvpB.

Independent groups have reported that *C. burnetii* lacking CvpB can still replicate to equivalent levels as wild-type bacteria in epithelial cells[9,29]. However, this is not the case in macrophages, where disruption of CvpB has a replication defect by 6 days post infection[28,31]. To comprehensively assess intracellular replication of the *cvpB*::Tn mutant in our laboratory, we conducted growth curves in both HeLa and THP-1 cells over a time course of infection, using different starting MOIs. In both cell types, *cvpB*::Tn showed delayed intracellular replication compared to wild-type (Supplementary Fig 3A, B). In THP-1 cells, equivalent levels were observed once replication reached a plateau. To assess the effect of bacterial load on cathepsin B, we therefore infected THP-1 cells with *cvpB*::Tn at MOIs of 25, 250 or 325. We observed a decrease in the amount of retained cathepsin B with increasing bacterial load, though it was not restored to wild-type levels (Supplementary Fig 3C, D), similar to what we observed for *icmL*::Tn (Fig. 1I).

To determine whether CvpB impacted the activity of other cathepsins, we utilised the BMV109 activity-based probe in the context of *cvpB*::Tn infection. No other cysteine proteases appeared to show altered activity when *cvpB* was disrupted (Fig. 3F), suggesting this phenotype is specific to cathepsin B. We also wanted to assess whether inhibition of cathepsin B activity would have any impact on *C. burnetii* replication. In this experiment, we chose to use a MOI of 0.1 so that we would observe a replication defect, which could theoretically be rescued by cathepsin B inhibition. Cells were treated with the specific inhibitor Ca074Me and intracellular replication was assessed over a time course. We observed that Ca074Me treatment did not alter replication dynamics of either wild-type or *cvpB*::Tn *C. burnetii* (Fig. 3G, H), indicating that the replication defect in *C. burnetii cvpB*::Tn is not a result of retained cathepsin B activity.

We next wanted to assess whether expression of CvpB alone was capable of modulating cathepsin B levels. When ectopically expressed, we observed partial co-localisation of 3xFLAG-CvpB with cathepsin B (Fig. 3I, J). However, there was no significant difference in overall cathepsin B abundance in the presence of 3xFLAG-CvpB compared to 3xFLAG alone (Fig. 3K). Co-immunoprecipitation experiments did not identify a direct physical interaction between CvpB and cathepsin B (Supplementary Fig 3E). Together, these data demonstrate that CvpB is necessary but not sufficient for cathepsin B removal during infection.

Though not biochemically characterised, CvpB has been shown to inhibit the phosphoinositide kinase PIKfyve, leading to enrichment of PI(3)P on the CCV membrane and contributing to the fusogenicity of the CCV[29]. To elucidate whether PIKfyve inhibition contributes to cathepsin B dynamics during infection, we utilised the PIKfyve inhibitor vacuolin-1. We hypothesised that if PIKfyve inhibition was responsible for cathepsin B loss, *cvpB*::Tn-infected cells would show reduced cathepsin B upon vacuolin-1 treatment. Consistent with the literature, vacuolin-1 induced lipidation of LC3-I to LC3-II in uninfected cells ([32,33], Supplementary Fig 3F). However, we observed no change to cathepsin B abundance in *cvpB*::Tn-infected cells treated with vacuolin-1 (Supplementary Fig 3F, G). This indicates that the involvement of CvpB in cathepsin B loss is independent of PIKfyve inhibition.

To exclude the role of the proteasome in CvpB-mediated cathepsin B depletion, we treated cells with the proteasome inhibitor MG132. After a 6 h treatment window, we were able to observe proteasome inhibition, evidenced by an accumulation of ubiquitin in treated cells (Supplementary Fig 3H). However, proteasome inhibition did not have any impact on the abundance of cathepsin B (Supplementary Fig 3H, I), demonstrating that the proteasome is not responsible for cathepsin B loss during infection.

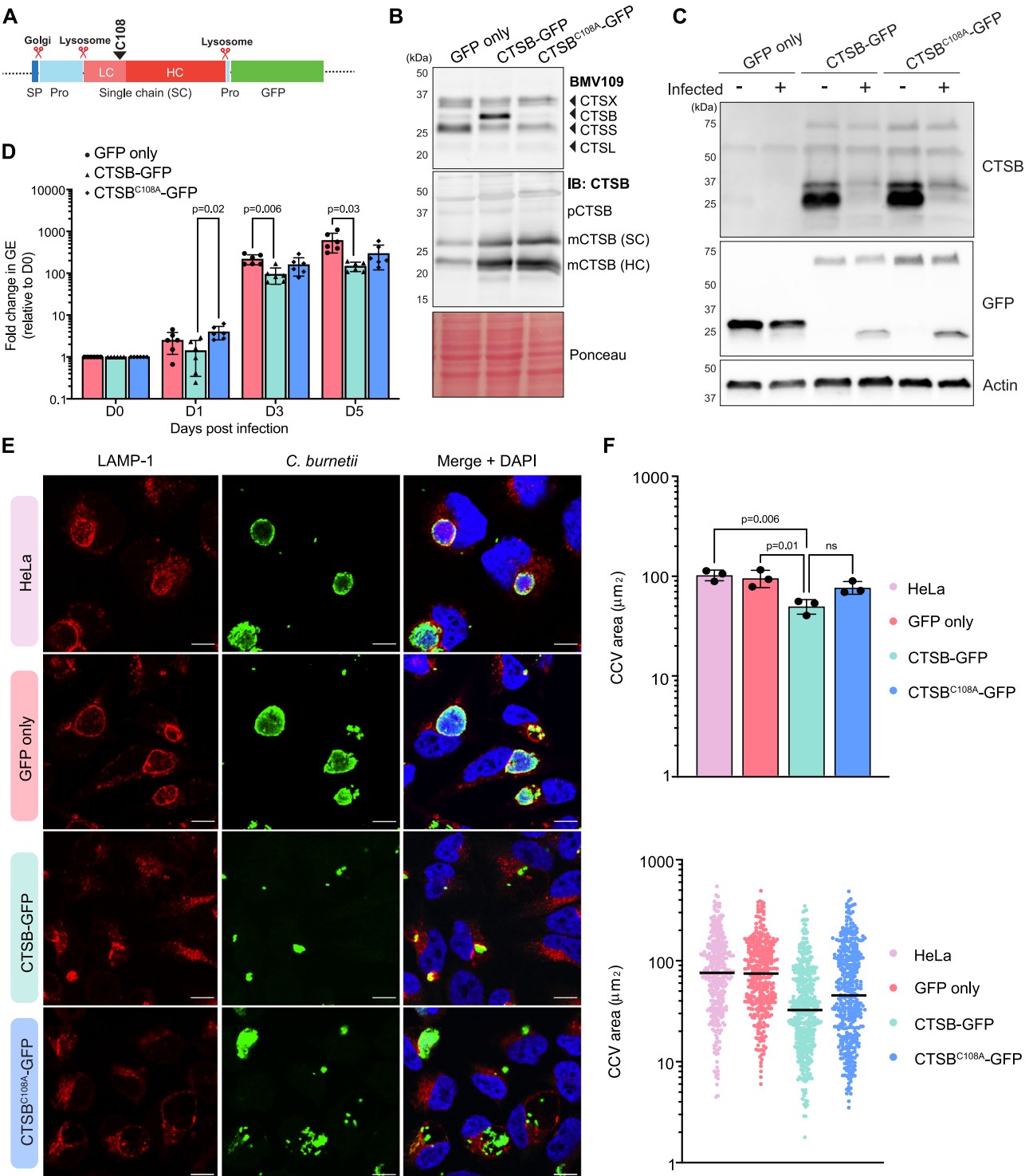

**Fig. 2 | Overexpression of cathepsin B is detrimental to *C. burnetii* replication and CCV biogenesis. A** Schematic depiction of cathepsin B-GFP domains and processing events. Catalytic cysteine is indicated above with black arrowhead. SP signal peptide, Pro pro-domain (**B**) HeLa cells or HeLa cells expressing GFP only, cathepsin B-GFP (CTSB-GFP) or catalytically inactive cathepsin B-GFP (CTSB^{C108A}-GFP) were labelled with BMV109 for 3 h prior to analysis of in-gel fluorescence (top) or immunoblotting for total cathepsin B (middle). Ponceau was used as a loading control (bottom). Pro pro-domain, SC single chain, HC heavy chain (**C–F**) HeLa, GFP, CTSB-GFP, or CTSB^{C108A}-GFP cells were uninfected or infected with *C. burnetii* at a MOI of 5. After 72 h, cells were harvested for immunoblotting (**C**), quantitative PCR (**D**), or fixed for immunofluorescence microscopy (**E, F**). **D** To quantify bacterial intracellular replication, cells were harvested immediately following infection

(D0), or at days 1, 3, 5, and 7 post-infection, genomic DNA was extracted, and *C. burnetii* genome equivalents quantified via qPCR with *C. burnetii* specific primers. Error bars represent standard deviation of six independent experiments that are represented by individual symbols. *P* values calculated by two-way ANOVA with Dunnett's correction for multiple comparisons. **E** Representative images of those used for quantification in **F**. **F** Quantification of CCV area as determined by ImageJ analysis of confocal images. Top panel, data points reflect average CCV area for each replicate, *n* = 3. For each replicate, >100 CCVs were measured and quantified. Statistical significance was determined using one-way ANOVA and comparing each group to CTSB-GFP. Error bars represent SD. Bottom panel, representation of spread of CCV areas from one replicate. Black line denotes mean. Scale bar = 10 μm. Source data are provided as a Source Data file.

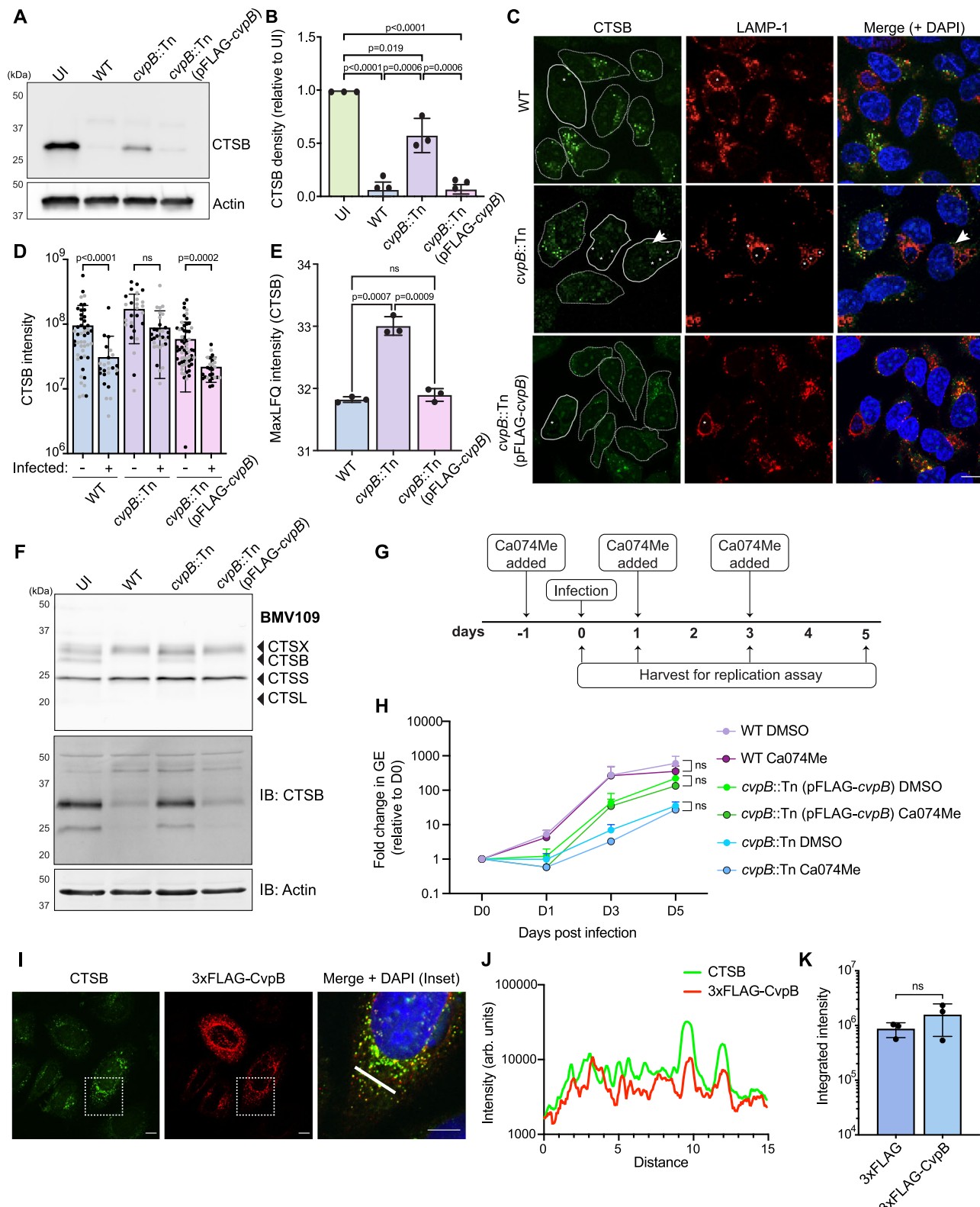

To date, the structure of CvpB has not been solved and low-confidence AlphaFold models limit the prediction of putative domains or binding regions. To identify the region(s) of CvpB that are required for cathepsin B loss, we generated a series of *C. burnetii cvpB*::Tn strains complemented with different N-terminal truncations, which all retain the C-terminal secretion signal required for translocation through the T4SS (Supplementary Fig 4A). None of these strains exhibited cathepsin B loss, suggesting that the N-terminus of CvpB is

required for cathepsin B removal (Supplementary Fig 4B, C). However, we also observed that none of these truncations were able to complement the multi-vacuole phenotype (Supplementary Fig 4D, E), confirming that the first 306 amino acids of CvpB are required for CCV fusogenicity as previously reported[29]. Taken together, these data give rise to the hypothesis that cathepsin B retention in cells infected with both *cvpB*::Tn or strains complemented with CvpB lacking its N-terminus is a result of different CCV environments compared to

**Fig. 3 | Loss of cathepsin B during *C. burnetii* infection requires the T4SS effector CvpB. A** Immunoblot of THP-1 cells uninfected (UI) or infected with *C. burnetii* wild-type (WT), *cvpB*::Tn or *cvpB*::Tn(pFLAG-*cvpB*) at MOI 25 for 72 h. Blot is representative of 3 independent experiments, quantified in **B**. Error bars denote SD. **C** Representative image of HeLa cells infected with the above strains and stained with antibodies against cathepsin B and LAMP-1. Scale bar = 10 μm. Dashed lines denote cell periphery of uninfected cells, solid lines, infected cells. CCVs indicated with an asterisk. **D** Quantification of CTSB signal intensity in **C**. Individual data points are from two independent experiments, coloured in black or grey, respectively. WT *n* = 25, *cvpB*::Tn *n* = 32, *cvpB*::Tn(pFLAG-*cvpB*) *n* = 27. Statistical significance determined using Kruskal–Wallis test. Data are presented as mean (column height), error bars represent SD. **E** Cathepsin B label-free quantification (LFQ) values from THP-1 cells infected with WT, *cvpB*::Tn, or *cvpB*::Tn (pFLAG-*cvpB*) (*n* = 3, MOI 50) for 72 h. Statistical significance was calculated using one-way ANOVA with Tukey's correction. Data are presented as mean (column height), error bars reflect SD. **F** BMV109 labelling (top) or immunoblot (bottom) on THP-1 cells infected with WT, *cvpB*::Tn or *cvpB*::Tn (pFLAG-*cvpB*) for 72 h (MOI 25). **G** Schematic depicting experimental design of data presented in **H**. **H** Analysis of bacterial intracellular replication in THP-1 cells infected with the indicated strains in the presence of DMSO or 10 μM Ca074Me. Data are presented as fold change relative to D0, with individual data points representing the mean of three biological replicates. Error bars denote SD. Statistical significance was examined using a paired *t*-test of each condition ± Ca074Me. **I–K** HeLa cells were transfected with pcDNA-3xFLAG-*cvpB* or pcDNA-3xFLAG for 18 h before immunofluorescence microscopy using antibodies against CTSB or FLAG. Scale bar = 10 μm. **J** Quantification of co-localisation in **I**. Area quantified is indicated in **I** by white line. **K** CTSB intensity in HeLa cells expressing 3xFLAG or 3xFLAG-CvpB. Data reflect the mean of three independent experiments (3xFLAG *n* = 70, 3xFLAG-CvpB *n* = 108). Statistical significance was examined using an unpaired *t*-test, error bars reflect SD. Source data are provided as a Source Data file. Mass spectrometry source data are available via ProteomeXchange with identifier PXD052954.

wild-type infection, and not a direct interaction of CvpB with cathepsin B.

## *C. burnetii* infection promotes extracellular secretion of lysosomal proteins, including cathepsin B

We next aimed to determine the cellular pathway behind cathepsin B loss during infection and queried whether the absence of intracellular cathepsin B was correlated with increased secretion of the protein into the extracellular space. Hypersecretion of lysosomal proteases is known to occur in several disease states, including cancer and inflammatory diseases[34]. Immunoblotting on conditioned media collected from infected cells revealed increased extracellular abundance of the pro-forms of cathepsins B and D (Fig. 4A, B). Equivalent amounts of actin were present in the media collected from uninfected and infected cells, confirming that differences in extracellular cathepsin abundance were not a consequence of differential cell lysis. To globally assess whether increased secretion was specific to cathepsins, we performed label-free proteomics on conditioned media collected from *C. burnetii*-infected THP-1 (Fig. 4C, D, F) or HeLa (Fig. 4E, F, Supplementary Data 1) cells. Cathepsin B was not identified to be statistically significantly increased at the protein level in these experiments due to variability in abundance of peptides in the pro or mature domains. However, a single peptide was identified from the pro-domain of cathepsin B, and this peptide was significantly increased in abundance during infection (Fig. 4C). In both cell types, gene ontology analysis revealed enrichment of proteins associated with lysosomes and vacuoles (Fig. 4F and Supplementary Table 1). We identified 6 proteins that displayed a statistically significant increase in extracellular abundance during infection of both THP-1 and HeLa cells (Fig. 4H, I), 5 of which are annotated to have lysosomal subcellular localisation in UniProt. These 6 proteins included hydrolases (CTSD, GLB1, PLBD2, ASAH1) and membrane proteins/receptors (LAMP2, ANXA5). Overall, these findings indicate that *C. burnetii* infection leads to a non-specific increase in the secretion of lysosomal content, which includes immature lysosomal hydrolases such as cathepsin B.

## Secretion of cathepsin B is independent of CvpB and involves the default secretory pathway

Given that CvpB modulates cathepsin B dynamics during infection, we assessed whether CvpB was also mediating the secretion of lysosomal proteins. PIKfyve inhibition has been shown to induce protein secretion via secretory autophagy or exosome release[35], so we examined the supernatant of cells infected with *cvpB*::Tn or *cvpB*::Tn (pFLAG-*cvpB*). Surprisingly, we saw no differences in the amount of cathepsin B in the extracellular media from wild-type and *cvpB*::Tn-infected cells (Fig. 5A, B). We deepened this analysis by performing proteomic analysis on supernatants and confirmed that the extracellular abundance of lysosomal proteins was not significantly altered when CvpB was disrupted, compared to WT or *cvpB*::Tn (pFLAG-*cvpB*) (Fig. 5C). This implies that while CvpB is involved in the intracellular loss of cathepsin B, this effector protein is not responsible for the global secretion of lysosomal content during infection.

We then assessed whether lysosomal protein secretion was dependent on the *C. burnetii* Dot/Icm T4SS. We observed equivalent cathepsin B secretion during infection with WT and *icmL*::Tn *C. burnetii* (Fig. 5D, E), which increased with MOI. Interestingly, this was completely uncoupled from intracellular cathepsin B abundance, which was consistently absent in WT infected cells but retained in *icmL*::Tn infected cells. This supports a model in which the intracellular cathepsin B loss is a distinct process from the global lysosomal protein secretion observed during infection. Furthermore, these data indicate that lysosomal protein secretion is likely a host response to infection, rather than an active pathogenesis mechanism employed by *C. burnetii*.

To uncover the cellular mechanism behind this secretory phenotype, we assessed whether enrichment of lysosomal proteins in the supernatant of infected cells was reflective of lysosomal exocytosis. This process involves recruitment of lysosomes to the cell surface and fusion with the plasma membrane, leading to release of lysosomal content into the extracellular space[36]. To investigate whether this occurs during *C. burnetii* infection, we immunostained live, unpermeabilised cells with an antibody to the lumenal epitope of LAMP-1[37]. During lysosomal exocytosis, this epitope is exposed on the cell surface and can be detected by confocal microscopy. As a positive control, cells were treated with the TRPML1 agonist ML-SA1, which resulted in a distinct LAMP-1 staining pattern at the cell periphery (Supplementary Fig 5A, top panel). This staining was not observed in untreated cells (Supplementary Fig 5A, middle panel) or in *C. burnetii*-infected cells (Supplementary Fig 5A, bottom panel). This suggests that the mechanism of lysosomal protein secretion during infection is distinct from lysosomal exocytosis.

It is now well established that disruption of the M6P trafficking pathway leads to hypersecretion of nascent lysosomal hydrolases[38–41]. Previous work found that *C. burnetii* could replicate in cells lacking GlcNAc1-phosphotransferase (GNPTAB−/−), which is responsible for adding an M6P to proteins destined for the lysosome[20], suggesting this pathway is not required for intracellular replication. However, it remains unknown whether the M6P trafficking machinery is functional during *C. burnetii* infection. We saw no difference in the abundance or subcellular distribution of the cation-independent-M6P receptor (CI-M6PR) as assessed by confocal microscopy (Supplementary Fig 5B). This, combined with the observation that other lysosomal proteins such as cathepsin D are still delivered to CCVs[9,14], suggest that M6P trafficking is functional during *C. burnetii* infection, and that secretion of lysosomal proteins is unlikely due to defects in the M6P machinery.

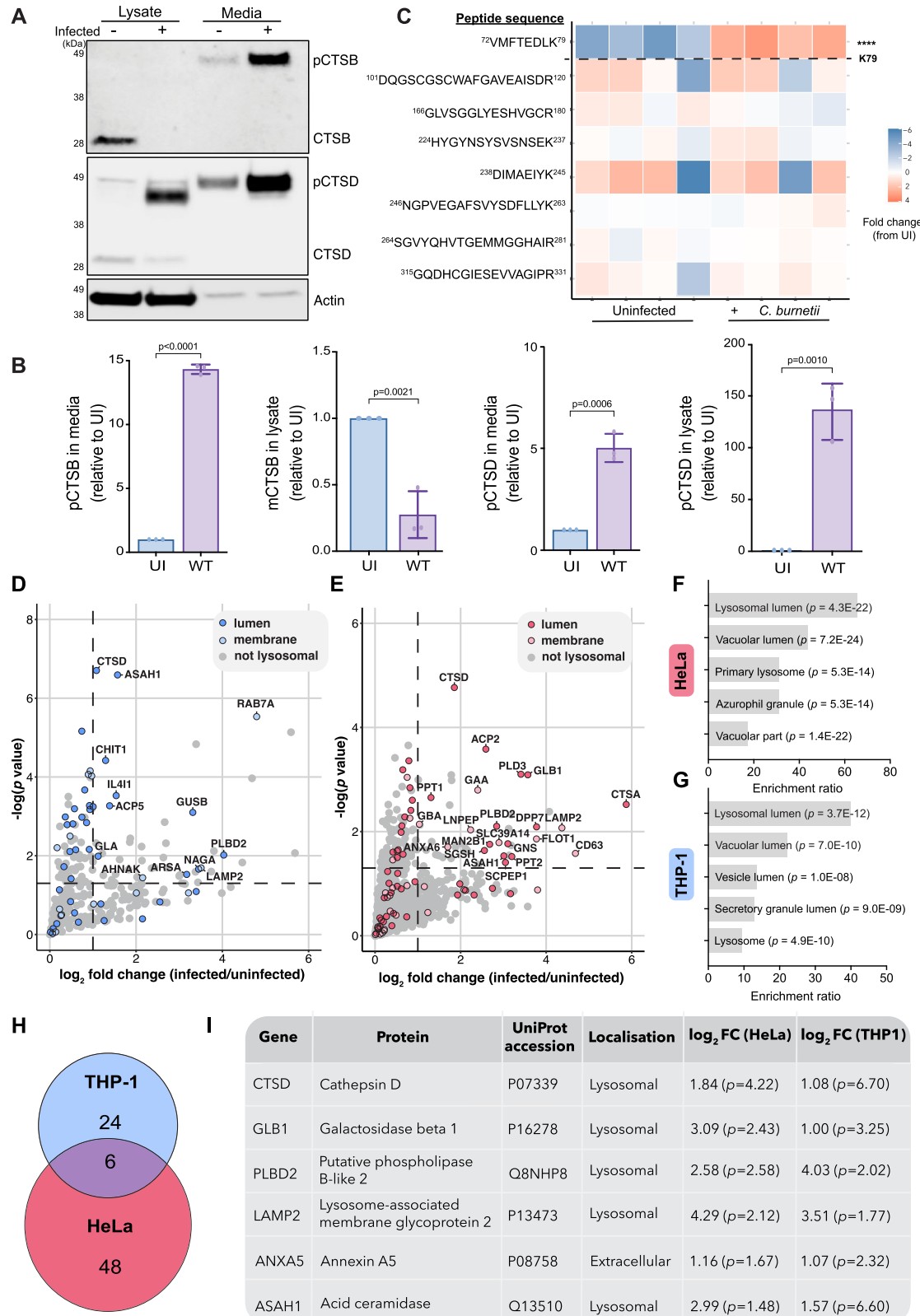

We then investigated whether we could pharmacologically inhibit the secretion of cathepsin B with brefeldin A (BFA), a classical inhibitor of the default secretory pathway from the Golgi apparatus to the plasma membrane. BFA inhibits vesicle formation between the ER and Golgi by inhibiting the small GTPase Arf1 and preventing COP-I coat recruitment[42,43]. Uninfected or *C. burnetii*-infected THP-1 cells were treated with 1 μM BFA for 6 h before both the media and cell fraction were collected for immunoblotting. BFA treatment had no significant effect on *C. burnetii* intracellular load as assessed by qPCR for determination of genome equivalents (Fig. 5F). As expected, BFA treatment led to accumulation of pro-cathepsin B in cell lysates, reflecting the zymogen form, which has not yet encountered a lysosomal environment for maturation. Furthermore, there was no change to the loss of cathepsin B in infected cells with the addition of BFA. Strikingly, BFA

**Fig. 4 | *C. burnetii* infection leads to the secretion of lysosomal content.**
**A** Immunoblot on whole cell lysates (lysate) or conditioned media (media) from uninfected or *C. burnetii*-infected HeLa cells. Cells were infected for 48 h (MOI 100) before being changed to serum-free media for a further 16 h. Blot is representative of 3 independent experiments, quantified in **B** Error bars denote SD.
**C–I** Conditioned media from uninfected ($n = 4$) or *C. burnetii*-infected ($n = 4$) THP-1 cells (**C**, **D**, **G**) or HeLa cells (**E**, **F**) were subject to label-free quantitative mass spectrometry. **C** Heatmap of MaxLFQ expression of peptides from cathepsin B, ordered from N-terminus (top) to C-terminus (bottom). Horizontal dashed line depicts end of pro-domain. Columns represent individual biological replicates.
**D**, **E** Scatter plot of proteins enriched in the supernatant of *C. burnetii*-infected cells, where *X*-axis represents fold change (infected/uninfected) and *Y*-axis shows statistical significance as determined by a student's *t* test. Proteins identified as located in the lysosome, lysosome lumen, or lysosome membrane (GO terms GO:0005764, GO:0043202, or GO:0005765, respectively) are coloured; all other proteins identified are depicted in grey. Horizontal dashed line denotes significance cut off ($p < 0.05$), vertical dashed line represents fold change > 1. Lysosomal proteins within these bounds are labelled with gene names. **F**, **G** GO analysis on significantly enriched proteins in **D**, **E** performed using WebGestalt overrepresentation test (Fisher's exact test with BH correction for multiple comparisons) and the cellular component functional database. **H** Venn diagram depicting overlap of significantly enriched proteins in the supernatant from THP-1 and HeLa cells. **I** Proteins identified as significantly enriched in the supernatant of infected THP-1 and HeLa cells. Significance was determined using a student's *t* test −log(*p* value) > 1.3 and log2 fold change > 1. Localisation is presented as designated in UniProt. Source data are provided as a Source Data file. Proteomics data have been deposited in ProteomeXchange with identifiers PXD052888 (THP-1) and PXD052890 (HeLa).

treatment completely abolished the secretion of pro-cathepsin B into the media. Taken together, this suggests that proteins are being secreted to the extracellular medium from the Golgi through the default secretory pathway.

Lastly, we aimed to determine whether the secretion of lysosomal content was important for *C. burnetii* infection. To assess this, we treated naïve cells with the secretome collected from either uninfected cells (mock) or *C. burnetii*-infected cells (Fig. 5I) and then assessed the intracellular replication ability of *C. burnetii*. We saw no measurable difference in replication (Fig. 5J), indicating that the secretion of lysosomal content does not change susceptibility to infection in cell culture conditions.

Collectively, we propose that two distinct processes are being subverted during *C. burnetii* infection. Firstly, a cohort of lysosomal proteins is secreted into the extracellular medium directly from the Golgi via the default secretory pathway. This secretion is not specific to cathepsin B but rather is observed for several lysosomal proteins, which show unchanged intracellular abundance, including cathepsin D. This is likely a host response to infection rather than an active pathogenesis mechanism employed by *C. burnetii*. Secondly, the pool of mature cathepsin B, which is retained inside cells, is removed by *C. burnetii* as a result of the vacuolar environment established by the Dot/Icm effector CvpB (Fig. 6).

## Discussion

While most intracellular pathogens have developed strategies to evade lysosomal degradation, *Coxiella burnetii*, the causative agent of Q fever, actively requires this environment for replication and virulence. Despite the established dogma that the *C. burnetii* replicative niche is proteolytically active, few studies have investigated the abundance or activity of lysosomal hydrolases present in the CCV[14,44]. We report that *C. burnetii*-infected cells exhibit loss of the lysosomal protease cathepsin B, a major workhorse of proteolysis in the lysosome (Fig. 1). Cathepsin B is a cysteine protease with broad substrate specificity, which is primarily involved in routine turnover of proteins in the lysosome. We hypothesise that *C. burnetii* removes cathepsin B to create a lysosomal environment that is less degradative and therefore more permissive of bacterial replication. Consistent with this, we observed that overexpression of cathepsin B had negative impacts on *C. burnetii* replication and CCV biogenesis (Fig. 2). During *C. burnetii* infection, the growing CCV is continuously damaged and repaired in an ESCRT-dependent manner[45]. Lysosome membrane permeabilization and release of cathepsin B into the cytosol can trigger activation of the NLRP3 inflammasome and subsequent pyroptotic cell death[46]. Removal of cathepsin B would therefore present an additional advantage in avoiding inflammatory cell death and clearance of infection.

We identified that cathepsin B is retained in cells infected with *C. burnetii* lacking the Dot/Icm effector CvpB, although still reduced compared to uninfected cells (Fig. 3). Although CvpB has been demonstrated to promote CCV biogenesis through inhibition of the phosphoinositide kinase PIKfyve[29], we determined that the role of CvpB in cathepsin B loss was not due to PIKfyve inhibition (Supplementary Fig 3F). CvpB is a 93 kDa multi-domain protein which remains incompletely characterised. It is conceivable that the relationship between CvpB and cathepsin B is therefore due to a novel function of this effector. Expression of CvpB alone was not sufficient to alter cathepsin B levels, and pulldown experiments did not reveal a physical interaction between the two proteins, suggesting an indirect interaction. It is interesting to note that *cvpB*::Tn-infected cells show a distinct multi-vacuole phenotype as a result of PI(3)P depletion[29,30]. This prevents fusion of CCVs with autophagosomes and other CCVs. The extent to which the pH and degradative properties of *cvpB*::Tn CCVs differ from wild-type is not fully established, although initial reports demonstrate that they retain the ability to hydrolyse DQ-BSA and stain positive for cathepsin D[9]. We therefore hypothesise that retention of cathepsin B during *cvpB*::Tn infection is a result of an altered CCV environment. CCV biogenesis, facilitated by CvpB, creates an environment within which cathepsin B is lost, likely due to either instability and/or other *Coxiella* factors having the opportunity to target this lysosomal protein for degradation. A multi-vacuole phenotype has also been observed in the absence of autophagy-related proteins (e.g. ATG5, STX17)[29,30,47]. However, it has never been demonstrated that the CCVs formed in the absence of autophagy have the same properties as CCVs formed in the absence of CvpB. Future studies should investigate this question and test whether retention of cathepsin B is directly linked to defects in homotypic fusion. Importantly, CvpB mutants display attenuated virulence in both *Galleria mellonella* and SCID mice[29,30,48]. Whether this is linked to cathepsin B retention is worthy of further consideration.

Our finding that cathepsin B is absent from cells infected with *C. burnetii* has important implications for the *C. burnetii* research field. Initial reports that the CCV is a proteolytically active compartment utilised the cathepsin B Magic Red substrate as a proxy for general proteolytic activity in *C. burnetii* infected cells[44]. However, using the covalent chemical probe BMV109, we demonstrate a consistent loss of cathepsin B activity during *C. burnetii* infection, while the activity of other cathepsins remains relatively constant. As there is considerable overlap in cathepsin cleavage specificity, combined with promiscuous cleavage activities of cathepsins and other proteases, substrate-based probes such as Magic Red often fail to discern the nuances of individual proteases. Our results, therefore, suggest that more cautious interpretations of Magic Red assays in the context of *C. burnetii* infections are warranted.

Why *C. burnetii* infection promotes the loss of cathepsin B over other cathepsins remains unclear. We did not observe a decrease in other proteases of the cathepsin family except for a slight reduction, but not complete absence, in cathepsin C expression at 3 days post infection (Fig. 1C). It is important to note that a comprehensive profile of cathepsin B substrates remains elusive, however proteomic

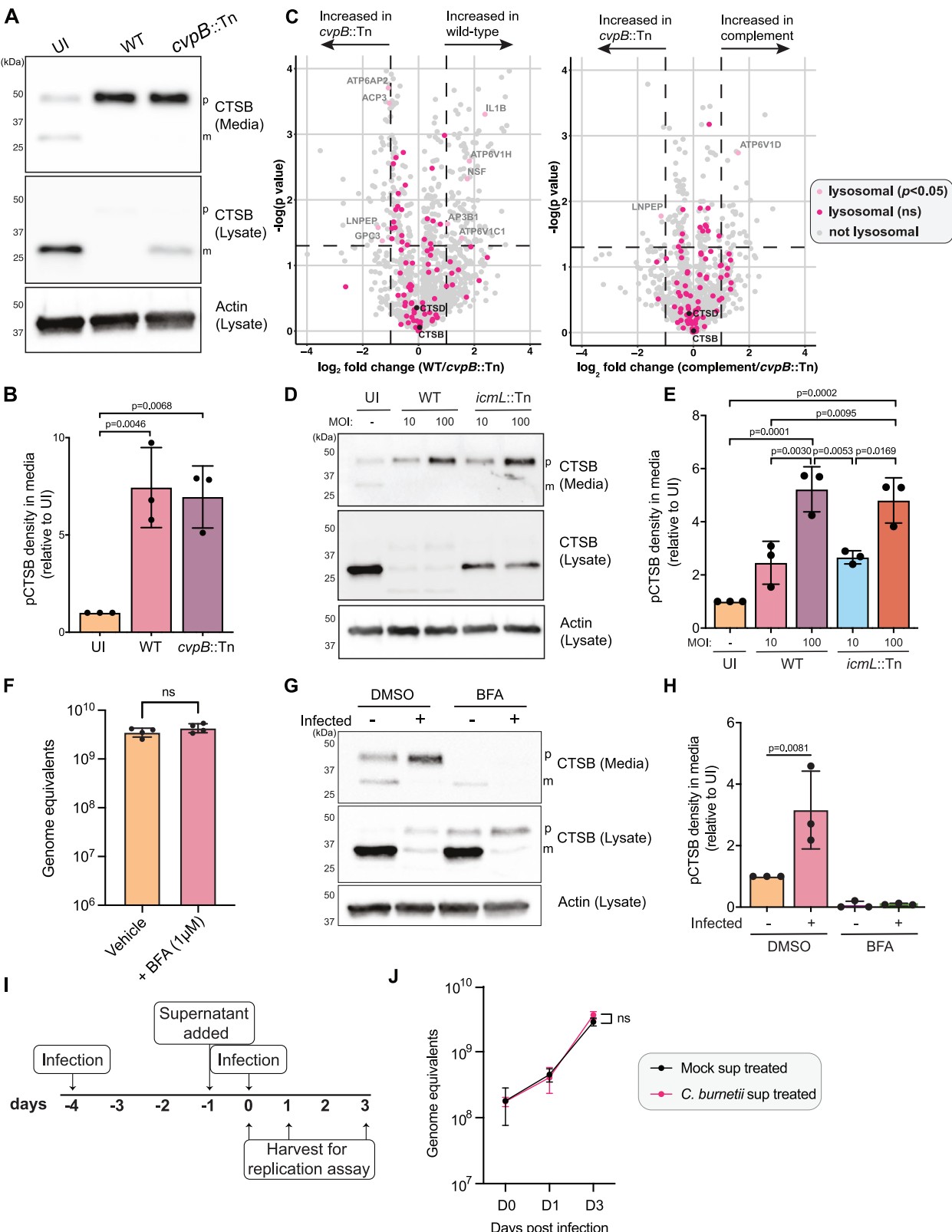

methods to identify carboxypeptidase substrates continue to improve and will likely provide an opportunity to elucidate the specific cathepsin B substrates that impact CCV biogenesis. This may aid in elucidating why cathepsin B is actively and selectively removed from *C. burnetii*-infected cells in contrast to other cathepsins. Interestingly, Qi et al. showed that genetic deletion of cathepsin B, but not cathepsin G or other non-lysosomal proteases, conferred enhanced intracellular

resistance to the cytosolic bacterium *Francisella novicida*[49]. The authors demonstrated that cathepsin B deletion promotes lysosomal biogenesis and autophagy by governing the activities of transcription factor EB (TFEB) and the autophagy kinase ULK1. While *C. burnetii* infection is known to promote engagement of the autophagic pathway[9,50], there is conflicting literature on whether TFEB is activated[21,51] or inhibited[52] during infection in a T4SS-dependent

**Fig. 5 | Secretion of lysosomal content is independent of CvpB and involves the default secretory pathway. A, B** Immunoblot on THP-1 lysates or media following infection with WT or *cvpB*::Tn. Blot is representative of 3 independent experiments, quantified in **B**. Statistical significance was determined by one-way ANOVA with Tukey's correction. Error bars denote SD. **C** Scatter plot of proteins enriched in the supernatant of *cvpB*::Tn ($n = 4$) or *cvpB*::Tn(pFLAG-*cvpB*) ($n = 3$) as determined by proteomics analysis. *X*-axis represents fold change (relative to WT), *Y*-axis shows statistical significance as determined by Student's *t* test. Proteins located in the lysosome, lysosome lumen, or lysosome membrane (GO terms GO:0005764, GO:0043202, or GO:0005765, respectively) are coloured; all other proteins identified are depicted in grey. Horizontal dashed line denotes significance ($p < 0.05$), vertical dashed line represents fold change >1. Lysosomal proteins within these bounds are labelled. **D** Immunoblot on THP-1 lysates or media following infection with WT or *icmL*::Tn at the indicated MOI. Blot is representative of 3 independent experiments, quantified in **E**. Mean is represented by column height, error bars represent SD. Statistical significance was determined using one-way ANOVA with Tukey's correction. **F** qPCR determination of *C. burnetii* genome equivalents after 48 h of infection followed by 6 h treatment with DMSO ($n = 4$) or 1 μM brefeldin A (BFA) ($n = 4$). Statistical significance was examined using an unpaired *t*-test. Error bars denote SD. **G, H** THP-1 cells were treated with BFA as in **F** for 6 h before conditioned media and lysate fractions were collected for immunoblotting. Data reflect three biological replicates, error bars denote SD. **I** Schematic of experimental design for data presented in **J. J** THP-1 cells were uninfected (mock, $n = 3$) or infected with WT ($n = 3$) for 72 h before the secretomes were collected and applied to naïve cells for 24 h. After this time, treated cells were infected and harvested for qPCR analysis at D0, 1, and 3 post infection. Data points represent the mean of three biological replicates. Statistical significance was examined using an unpaired *t*-test at D3 post-infection. Error bars denote standard deviation. All source data are provided as a Source Data file. Mass spectrometry data have been deposited in ProteomeXchange with identifier PXD052954.

manner. It is likely that the true nature of TFEB signalling during infection is dynamic and influenced by multiple factors (as reviewed in ref. [5]). Nonetheless, it is plausible that the reported increase in lysosomal biogenesis and autophagy observed during *C. burnetii* infection could be a downstream effect of the cathepsin B loss we observe in our study.

Our second major observation is that *C. burnetii* infection causes a suite of lysosomal proteins to be secreted from the Golgi apparatus to the extracellular space, which can be inhibited by Brefeldin A (BFA) and is independent of CvpB activity (Figs. [4] and [5]). Although we observed enrichment of lysosomal proteins in the media, we did not detect LAMP-1 on the cell surface during infection, suggesting this was not reflective of lysosomal exocytosis. It was recently shown that β-coronaviruses incite lysosomal exocytosis for egress, and the authors determined that this pathway is insensitive to BFA[53]. In contrast, our findings show that lysosomal protein secretion could be inhibited by BFA, suggesting a distinct cellular pathway is engaged during *C. burnetii* infection.

We found that secretion of lysosomal proteins occurred even during infection with a *C. burnetii* Dot/Icm mutant (Fig. [5]D, E) and was exacerbated with increasing bacterial load. We therefore propose that the secretion may be a compensatory mechanism by the host cell to maintain homoeostasis following infection. The occupation and modulation of host lysosomes by *C. burnetii* is known to globally alter the degradative capacity of the cell[18,26], and reports have demonstrated that pathways such as secretory autophagy and exosome release can be incited as an alternative waste disposal pathway to maintain homoeostasis when lysosomal degradation is impaired[54–56]. Elucidating the mechanics of this pathway further will be important for a deeper understanding of the molecular mechanisms subverted during *C. burnetii* infection.

Redirection of lysosomal hydrolases is not unique to *C. burnetii* infection. *Helicobacter pylori* was recently shown to induce secretion of the lysosomal protease legumain to promote cleavage of extracellular matrix proteins and subsequent tumour invasion in a gastric cancer model[57]. *Salmonella enterica* Typhimurium encodes a T3SS effector, SifA, which reduces the hydrolytic capacity of lysosomes by interfering with retrograde trafficking of M6PR, causing secretion of hydrolases to the extracellular space[58]. Further research demonstrated that in addition to being secreted, cathepsins are trafficked to the nucleus during *S.* Typhimurium infection, where they contribute to non-canonical inflammasome activation and cell death[59]. While we demonstrate that treatment of naïve cells with the secretome from *C. burnetii* infected cells had no impact on intracellular replication of *C. burnetii* under the in vitro conditions tested (Fig. [5]J), it should be investigated whether inflammatory pathways or the transcriptomic landscape of neighbouring cells is affected. Within cell culture conditions, these secreted proteases are in their zymogen form but in the context of infected tissue and inflammatory conditions, this may not be the case and thus it remains plausible that secretion of lysosomal enzymes influences the host response to infection.

Overall, this study has established that lysosomal proteolysis and trafficking are actively modulated during *C. burnetii* infection, through both removal of mature cathepsin B from infected cells and secretion of a cohort of lysosomal hydrolases to extracellular space. Taken together, these findings contribute significantly to our understanding of how this unique intracellular bacterium subverts the host lysosome to cause disease.

## Methods

### Bacterial strains, plasmids and oligonucleotides
*C. burnetii* Nine Mile Phase II RSA439 were grown axenically in ACCM-2 liquid cultures at 37 °C with 5% $CO_2$, and 2.5% $O_2$ for 6–7 days[60]. For growth of mutants, media were supplemented with 350 μg/mL kanamycin or 3 μg/mL chloramphenicol as required. Strains used in this study are detailed in Supplementary Data 3. *Escherichia coli* pir2 or Stbl3 strains were cultivated in Luria-Bertani (LB) broth or LB agar plates supplemented with kanamycin (50 μg/mL), ampicillin (100 μg/mL), or chloramphenicol (12.5 μg/mL) as required. Plasmids used in this study are also detailed in Supplementary Data 3.

Custom oligonucleotides were ordered through Sigma Aldrich or Integrated DNA Technologies (IDT). A table of oligonucleotides used in this study can be found in Supplementary Data 3.

### Cell culture
HeLa CCL2 cells (ATCC) were routinely passaged in Dulbecco's Modified Eagle Media + GlutaMAX (DMEM) (Gibco, Thermo Fisher) supplemented with 10% heat-inactivated foetal calf serum (FCS). THP-1 human monocytic cells were maintained in Roswell Park Memorial Institute (RPMI) media with 10% FCS and differentiated with phorbol 12-myristate 13-acetate (PMA) at a concentration of $10^{-8}$ M for 72 h prior to infection. All cell lines were maintained at 37 °C with 5% $CO_2$.

### Generation of stable cell lines
HEK293T cells were transfected with lentiviral packaging plasmids alongside a lentiviral expression vector containing the protein of interest (see Supplementary Data 3). After 24 h, viral supernatant was collected and filtered through a 0.45 μm filter. Polybrene was added to a final concentration of 8 μg/ml before viral titres were added to HeLa cells in a 6-well plate and centrifuged for 1 h at 1583 × *g* at 32 °C. After 48 h, cells were incubated in media containing 5 μg/ml puromycin to select for successfully transduced cells. Expression of the desired protein was confirmed by WB.

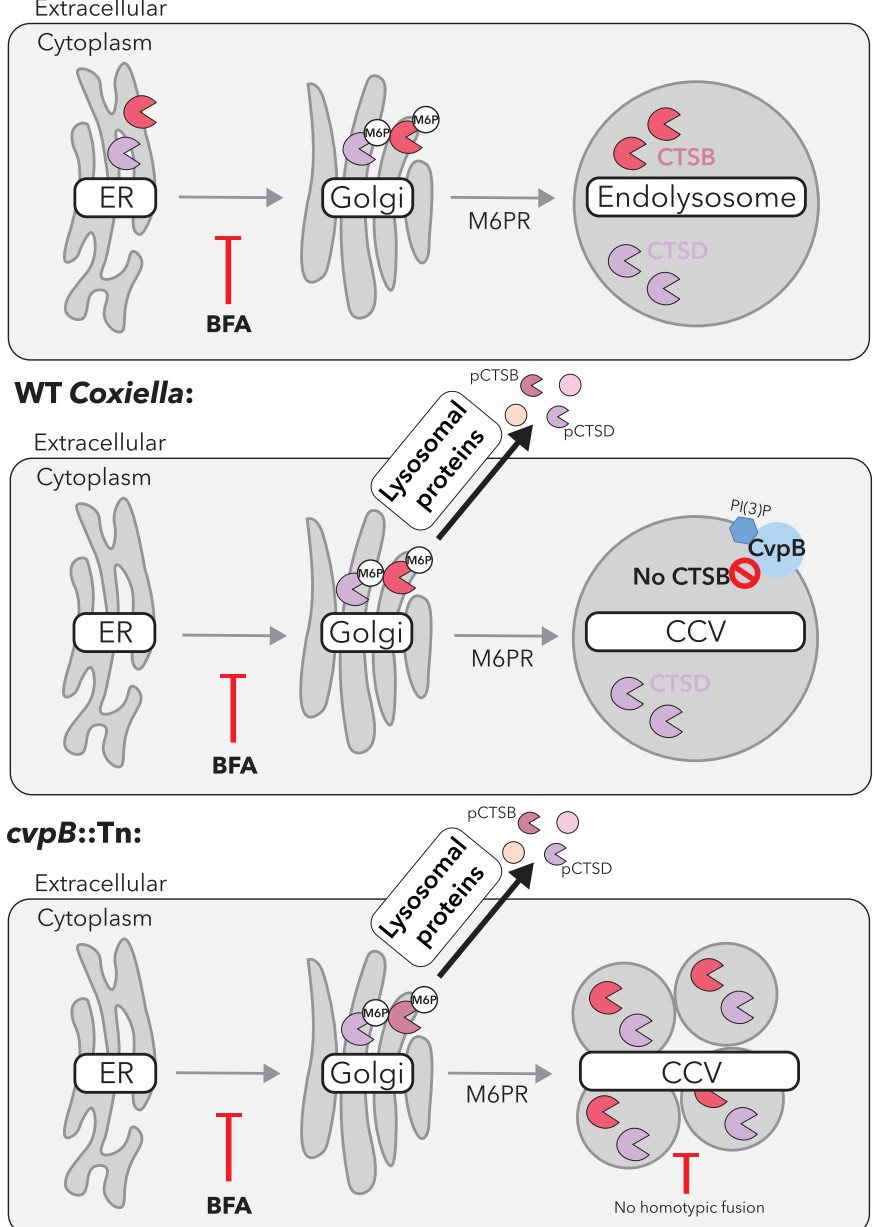

**Fig. 6 | *C. burnetii* subverts the host lysosome and trafficking pathways to facilitate intracellular success.** Proposed model for the response of mammalian cells to infection with WT or *cvpB*::Tn *C. burnetii*. Infection with both strains leads to secretion of a cohort of lysosomal proteins to the extracellular space through a mechanism that can be inhibited with BFA. Infection with WT, but not *cvpB*::Tn leads to removal of cathepsin B (CTSB) from the *Coxiella*-containing vacuole (CCV). Figure created in BioRender. Bird, L. (2025) https://BioRender.com/c7kaafo.

### *C. burnetii* infection of cultured cell lines

HeLa or differentiated THP-1 cells were infected with *C. burnetii* as described previously[61]. Briefly, *C. burnetii* genome equivalents from stationary phase cultures were quantified by RT-qPCR using primers specific for *ompA*. *C. burnetii* was added to cells at an MOI of 100, 50, 5 or 1, depending on the cell type and experimental design. Infected cells were centrifuged at $500 \times g$ for 30 min or incubated at 37 °C with 5% $CO_2$ for 4 h before washing once with phosphate-buffered saline (PBS) and once with respective cell culture media (DMEM or RPMI) supplemented with 5% FCS.

### Antibodies and reagents

The following antibodies were used for WB or IF: anti-cathepsin B monoclonal (Cell Signalling Technology #31718, WB 1:2000, IF: 1:200), anti-cathepsin B polyclonal (R&D Systems, #AF965, WB 1:1000), anti-cathepsin C (Santa Cruz Biotechnology #sc-74590, WB 1:2000), anti-cathepsin D (Abcam #ab72915, WB 1:2000), anti-DotB (Edward Shaw Laboratory, WB 1:2000), anti-β-actin (Sigma Aldrich #A1978, WB 1:8000), anti-3xFLAG (Sigma, #F3165, IF 1:250) anti-LAMP-1 (Developmental Studies Hybridoma Bank #H4A3, IF 1:250), anti-*Coxiella burnetii* (Craig Roy Laboratory, IF 1:10,000), anti-LC3B (Novus, #NB100-2220, WB 1:2000), anti-CI-M6PR (Novus #NB300-514, IF 1:100), anti-Ubiquitin (Cell Signalling Technology #3936S, WB 1:2000).

Other reagents used in this study include brefeldin A (Invitrogen #00-4506-51), ML-SA1 (Sigma Aldrich #SML0627), Ca074Me (Sigma Aldrich #C5732)[62], MG-132 (Sigma Aldrich #474790), and vacuolin-1 (Sigma Aldrich #673000). BMV109 was synthesised as described in refs. 24,25.

## Western blotting

For analysis of protein expression in whole cell lysates, mammalian cells were infected with *C. burnetii* as above. At 3 dpi, cell lysates were harvested in 2X Laemmli sample buffer for SDS-PAGE. For examination of proteins in cell supernatants, media from cells were collected after 48 h of infection and centrifuged at $21,000 \times g$ for 5 min to pellet cell debris. Supernatant was collected and centrifuged through Amicon® Ultra Centrifugal Filters with 10 kDa molecular weight cut-off (Merck). The concentrated supernatant was combined with 2X Laemmli sample buffer for SDS-PAGE. All samples were boiled at 95 °C prior to SDS-PAGE, then electrophoresed at 165 V for 45 min on an Any kD™ Mini-PROTEAN® TGX Stain-Free™ protein gel (Bio-Rad), before being transferred to PVDF membranes and blocked for 1 h with Tris-buffered saline plus 0.1% Tween-20 (TBST) containing 5% skim milk. Primary and secondary antibodies (see above) were diluted in TBST containing 5% bovine-serum albumin (BSA). Chemiluminescence was detected using the Bio-Rad Clarity Western WCL kit using an Amersham 680 Imager (GE Healthcare) or Bio-Rad ChemiDoc MP Imager. Where indicated, BMV109 (1 µM) was added to cells 4 h prior to harvesting, and cathepsin activity was visualised by in-gel fluorescence by scanning with a Cy5 filter on a Typhoon 5 (GE Healthcare). All densitometry was performed using ImageLab software (BioRad). Intensity values were expressed as a percentage of the uninfected control for each experiment and then normalised against the respective expression of actin. Uncropped western blots are available in Source data.

## qRT-PCR

Differentiated THP-1 cells were infected for 3 days with *C. burnetii* (see above). RNA extraction was performed using a phenol-chloroform method as previously described[63]. Briefly, cells were lysed in TriSURE™ (Bioline) and incubated with chloroform to achieve phase separation. The RNA-containing layer was isolated and precipitated with isopropanol before being washed and quantified using a Nanodrop 2000 Spectrophotometer (ThermoFisher). Any contaminating genomic DNA was removed via rDNase I treatment, followed by addition of DNase inhibitor (Ambion, USA). RNA was reverse transcribed to cDNA using the iScript™ cDNA synthesis kit (BioRad). RT-qPCR was performed using SYBR Green reaction mix (BioRad) and oligonucleotides listed in Supplementary Data 3. Real-time PCR was performed using a QuantStudio™ 7 Flex Real-Time PCR system (ThermoFisher). Transcript abundance for the RNA of interest was normalised against 18S rRNA abundance.

## Immunofluorescence microscopy

HeLa cells were infected for 3 days with *C. burnetii* on 12 mm glass coverslips in 24-well plates. At 3 dpi, cells were fixed with 4% (w/v) paraformaldehyde for 15 min at room temperature. Fixed cells were permeabilised with 0.1% Triton X-100 in PBS at room temperature for 20 min before blocking in PBS containing 1% (w/v) bovine serum albumin and 0.01% Tween-20 for 1 h (blocking buffer). Stable cell lines expressing GFP were photo-bleached prior to immunostaining to quench endogenous GFP signal. Immunostaining was performed with antibodies diluted in blocking buffer. Prior to mounting onto glass slides, coverslips were incubated with PBS containing 4′, 6′-Diamidino-2-phenylindole (DAPI, Life Technologies) at a dilution of 1:10,000. Images were acquired using a Zeiss LSM780 confocal microscope using a 63X oil objective lens (Zeiss, Germany). Image processing was performed in ImageJ.

## Quantification of CTSB signal intensity

Quantification of CTSB in immunofluorescence images of transfected and infected cells was performed with CellProfiler[64]. Briefly, cells were manually outlined using FLAG (for transfected cells) or CTSB (for infected cells) staining, and background signal was subtracted using the mean intensity of a cell-free area. Nuclei were automatically identified from DAPI staining and masked, and CTSB puncta were identified using Otsu's thresholding method. The integrated density of CTSB puncta was measured and combined per cell. Additionally, for infection experiments, CCVs were manually identified using DAPI staining and used to classify cells as either infected or uninfected.

## LAMP-1 cell surface staining

To detect LAMP-1 on the plasma membrane, 3-day infected HeLa cells were washed thrice with warm PBS before being treated with 50 µM ML-SA1 in DMEM + 5% FCS for 5 min. After treatment, cells were transferred to a metal plate on ice to prevent re-endocytosis of LAMP-1. All subsequent procedures were performed at 4 °C. Live cells were stained with an antibody to the lumenal epitope of LAMP-1 (DSHB, 1:250) diluted in ice-cold PBS for 45 min on ice. Following this, cells were washed three times and then fixed with 4% PFA. Secondary antibody, DAPI staining, and image acquisition were performed as above.

## Label-free quantitative mass spectrometry

**Sample preparation.** HeLa or THP-1 cells were seeded in 10 cm dishes at a density of $1.1 \times 10^6$ or $1.1 \times 10^7$ cells per dish, respectively. Cells were infected with *C. burnetii* as above, with n = 4 biological replicates for each condition. After 48 h of infection, culture media were removed and replaced with fresh media containing no FCS (serum-free), and cells were incubated in this medium for 18 h. Conditioned media were collected and centrifuged at $3000 \times g$ for 10 min to pellet any cell debris. The supernatant was then concentrated through Amicon® Ultra Centrifugal Filters with a 10 kDa molecular weight cut off (Merck). The concentrated supernatant was combined with a lysis buffer containing 10% sodium dodecyl sulphate (SDS), 100 mM Tris-HCl pH 8. Samples were boiled at 95 °C on a thermomixer with shaking at 2000 rpm to shear DNA before the protein content was quantified using a BCA assay (Pierce). One hundred micrograms protein was used for mass spectrometry sample preparation. Briefly, samples were reduced with dithiothreitol (DTT) to a final concentration of 10 mM at 95 °C for 10 min, then alkylated with 40 mM iodoacetamide for 30 min at room temperature in the dark. The reaction was quenched by the addition of another 10 mM DTT, then samples were acidified with 12% phosphoric acid. Samples were combined with a binding/wash buffer containing 100 mM Tris pH 8, 90% methanol and loaded onto S-trap mini columns (ProtiFi, USA). Columns were washed 4x with binding/wash buffer and then on-column digestion was performed using 1 µg trypsin per 33 µg protein diluted in 50 mM Tris-HCl pH 8. Digestion was performed overnight at 37 °C in a humified container. The following day, samples were eluted via sequential centrifugation using 50 mM Tris-HCl pH, followed by 0.2% formic acid, then 50% acetonitrile. Each spin was $4000 \times g$ and the flow-through from each was collected. Eluate was vacuum dried before being resuspended in mass spectrometry running buffer (0.1% trifluoroacetic acid, 2% acetonitrile) and subjected to clean up on $C_{18}$ stage tips[65,66].

## Reverse phase liquid chromatography–mass spectrometry

$C_{18}$ enriched proteome samples were re-suspended in mass spectrometry running buffer and separated using a two-column chromatography setup composed of a PepMap100 $C_{18}$ 20-mm by 75-mm trap (Thermo Fisher Scientific) and a PepMap $C_{18}$ 500-mm by 75-mm analytical column (Thermo Fisher Scientific) using a Dionex Ultimate 3000 UPLC (Thermo Fisher Scientific). Samples were concentrated onto the trap column at 5 µl/min for 6 min with Buffer A (0.1% formic acid, 2% DMSO) and then infused into an Orbitrap Lumos™, Orbitrap Eclipse™ or Q-Exactive (Thermo Fisher Scientific) at 300 nl/min via the analytical columns. Peptides were separated by altering the buffer composition from 3% Buffer B (0.1% formic acid, 77.9% acetonitrile, 2% DMSO) to 23% B over 89 min, then from 23% B to 40% B over 10 min and then

from 40% B to 80% B over 5 min. The composition was held at 80% B for 5 min before being returned to 3% B for 10 min. The mass spectrometer was operated in a data-dependent mode automatically switching between the acquisition of a single Orbitrap MS scan (300–2000 m/z, maximal injection time of 50 ms, an Automated Gain Control (AGC) set to a maximum of 400k and a resolution of 60k) and 3 s of Orbitrap MS/MS HCD scans of precursors (NCE 35%, a maximal injection time of 80 ms, a AGC of 125k and a resolution of 30k).

## Data independent acquisition-mass spectrometry

Purified peptide samples were re-suspended in Buffer A* and separated using a trap and elute setup with a Pepmap nano-trap column (C18, 100 Å, 75 µm × 2 cm) and 5.5 cm high throughput uPAC Neo analytical column (Thermo Scientific) on a Vanquish Neo UHPLC (Thermo Scientific) coupled to an Orbitrap Astral mass spectrometer (Thermo Scientific). Peptides were separated using a gradient composed of 0.1% v/v formic acid (solvent A) and 80% acetonitrile with 0.1% v/v formic acid (solvent B). A flow rate of 750 nl/min was used with a gradient of (i) 0–0.3 min 3–6% B, (ii) 0.3–23 min, 6–23.5% B (iii) 23–26.7 min, 23.5–40% B (iv) 26.7–28.7 min 40–50% B and (v) 28.7–28.8 min, 50–99% B. Prior to sample loading, the analytical column was washed at 2 µl/min (combine flow and pressure control) for 14 column volumes, then the trap column was washed (combine flow and pressure control) with 2 zebra wash cycles at 80% acetonitrile. For DIA experiments, MS1 scans covering the range 380–980 m/z were acquired in the orbitrap at a resolution of 120,000 at m/z 200 between with an AGC target was 500% with a maximum IT of 5 ms in profile mode. MS2 scans were carried out in the Astral analyser with isolation window of 2 m/z, normalised HCD collision energy of 27, normalised AGC target of 500% and maximum injection time of 3 ms.

## Data analysis

Raw data files were searched in FragPipe (v.18.0)[67,68] against *Homo sapiens* (UniProt accession: UP000005640, downloaded 2023, 82518 entries) and *Coxiella burnetii* RSA439 NMII (UniProt accession: UP000002671, downloaded 2023, 1812 entries) reference proteomes. FragPipe parameters were set to default unless otherwise specified. Label-free quantification (LFQ) was undertaken allowing for the following modifications: cysteine carbamidomethylation as a fixed modification (+57.0215 Da) and methionine oxidation (+15.9949 Da) and N-terminal acetylation (+42.0106 Da) as variable modifications. Enzyme cleavage specificity was set to 'stricttrypsin', cleavage at K or R, allowing a maximum of 2 missed cleavages. Protein and peptide FDR were calculated using Philosopher with default settings. DIA datasets were searched using DIA library-free analysis within DIA-NN (version 1.8.1)[69]. Data files were searched against the *Homo sapiens* and *Coxiella burnetii* RSA439 NMII proteomes (see above for UniProt identifiers). Fixed and variable modifications, and protease specificity were defined as outlined above. For all datasets, the resulting data were further analysed using Perseus (v.1.6.0.7)[70]. A log$_2$ transformation was applied and data were filtered to contain valid values in a minimum of three out of four biological replicates in at least one group (i.e. uninfected/infected). Imputation was performed to replace missing values based off a downshifted normal distribution ($\sigma$-width = 0.3, $\sigma$-downshift = −2.3). Statistical comparison between groups was performed using a student's *t* test with a multiple hypothesis correction undertaken using a permutation-based FDR. Downstream data visualisation was performed using the ggplot2 package in R (v.4.2.1).

## Statistical analysis

Calculation of statistically significant differences between groups was performed in GraphPad Prism 9 (GraphPad Inc., USA). Where required, a one-way or two-way analysis of variance (ANOVA) with Tukey's post-hoc test was performed with $p < 0.05$ as a significance threshold, assuming equal standard variance between groups.

For mass spectrometry data, log$_2$ fold change in protein abundance between groups was calculated using a two-sided *t*-test. To adjust for multiple hypothesis testing, a Benjamini-Hochberg correction was performed using a false discovery rate (FDR) of 5%. Enrichment analysis was undertaken using WebGestalt[71,72] on significantly enriched proteins (−log($p$ value) > 1.3 and log$_2$ fold change >1) using the gene ontology cellular component functional database and the *H. sapiens* protein-coding genome as background.

## Reporting summary

Further information on research design is available in the Nature Portfolio Reporting Summary linked to this article.

## Data availability

The mass spectrometry proteomics data have been deposited to the ProteomeXchange Consortium via the PRIDE[73] partner repository with the dataset identifiers PXD052888, PXD052890 and PXD052954. Source data are provided with this paper.

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

## Acknowledgements

We wish to thank the Biological Optical Microscopy Platform (BOMP) and the Mass Spectrometry and Proteomics Facility (MSPF) at the University of Melbourne for access to and maintenance of key resources. L.E.B., A.R.Z., and B.X. are supported by RTP scholarships from the Australian Government. L.E.E.M. was funded by a grant from the Australian National Health and Medical Research Council (GNT2011119). Research in H.J.N.'s laboratory was funded by the Australian National Health and Medical Research Council (GNT2010841).

## Author contributions

L.E.B.: Conceptualisation, experimental investigation, data analysis and visualisation, writing—first draft, review and editing. B.X., A.D.H., A.R.Z., P.N., and D.R.T.: Experimental investigation. N.E.S. Experimental investigation—mass spectrometry and data analysis. L.E.E.M., and H.J.N.: Conceptualisation, experimental investigation, review and editing, supervision, funding acquisition.

## Competing interests

The authors declare no competing interests.
