## [Transparent Peer Review file · Nature Communications]

***Coxiella burnetii* manipulates the lysosomal protease cathepsin B to facilitate intracellular success**

Corresponding Author: Professor Hayley Newton

Version 0:

Reviewer comments:

Reviewer #1

(Remarks to the Author)

In Bird et al, the authors examine manipulation of host lysosomal proteases by the obligate intracellular bacterial pathogen *Coxiella burnetii*. *Coxiella* survives in a modified phagolysosome (aka CCV) known to contain lysosomal hydrolases and proteases, yet the bacteria avoids degradation. This study further examines the role of host proteases and how *Coxiella* may be manipulating them during infection. The authors convincingly show that intracellular cathepsinB protein levels and activity are decreased in *Coxiella*-infected cells. Overexpression of cathepsinB has a modest effect on *Coxiella* replication and CCV size. The *Coxiella* secreted effector protein CvpB is implicated in manipulating cathepsinB levels, albeit through an unidentified mechanism. Finally, the authors discovered that secretion of lysosomal proteases is increased in *Coxiella*-infected cells, in a cvpB-independent manner. The resulting model proposes that *Coxiella* infection leads to increased secretion of lysosomal proteins to the extracellular space, while cvpB is involved in selectively removing cathepsinB from the CCV. Overall, the paper is well written, figures clearly presented and the experiments well-executed. The finding that *Coxiella* increases secretion of lysosomal proteins provides important insight into *Coxiella* infection. However, significant gaps remain, including mechanism for cvpB and evidence that the proposed model is critical for successful *Coxiella* infection.

Decreased cathepsinB in *Coxiella*-infected cells (Figure1):

The data is convincing that, in *Coxiella*-infected cells compared to uninfected cells, pro-CathepsinB is secreted to the extracellular space while there is a dramatic decrease in catalytically-active cathepsinB. While this is interpreted as cathepsinB is removed by the bacteria (specifically cvpB), it is also plausible *Coxiella* blocks general cathepsin processing through an indirect mechanism. Supporting this, in Figure 1C and 3A there appears to be an increase in pro-cathepsinD in infected cells suggesting a defect in cathepsinD processing; it's less clear whether cathepsinB processing is being altered as pro-cathepsinB is not shown, although in Figure 5C there does appear to be pro-cathepsinB in infected by not uninfected DMSO-treated cells. Others studies have demonstrated that *Coxiella*-infected cells have few lysosomes and an expanded pool of less acidic endosomes (PMID: 28293541 and PMID: 31869379), which could lead to a defect in cathepsin processing. The authors should test whether decreased levels of proteolytically active cathepsinB is due to the altered endosomal compartment in *Coxiella*-infected cells; one possible experiment is overexpression of TFEB to increase lysosomal biogenesis. It would also be helpful to clarify whether both pro- and processed forms of cathepsinB are being detected in the mass spec, which portions the antibodies recognize for IFA and immunoblotting, and where the qRT-PCR primers are targeted.

The representative image in Figure 1F is dramatic and impressive. Does the antibody used recognize only processed cathepsinB? Is there any cathepsinB in the cell? This data would be strengthened by quantitating cathepsinB signal in uninfected and infected cells for WT as well as icmL and icmS mutants. This is true of other figures, and both microscopy and immunoblotting should be quantitated whenever possible.

Overexpression of CTSB is detrimental to *Coxiella* (Figure 2):

Figure 2D and line 175: The catalytically-inactive CTSB leads to an intermediate phenotype, as there is no significant difference between the catalytically inactive CTSB and either GFP or active CTSB. Thus, the statement that *Coxiella* "replication was lower in CTSB-GFP cells than cells expressing GFP alone...this was restored when cathepsinB was catalytically inactive" is overinterpreted. This intermediate phenotype is also seen in CCV size. The intermediate phenotype of the catalytically inactive CTSB, especially at 5 dpi, suggests that there might be a time-dependent role for CTSB during

infection. Measuring viable bacteria using colony forming unit (CFU) assays instead of genome equivalents would generate more accurate insight into the effect of cathepsinB on Coxiella.

Figure 2: Please clarify SC and HC in the figure legend (2B) and indicate where the processing events occur (2A).

2B and 2C: Have GFP immunoblots been performed, instead of total CTSB? This might provide insight into CTSB processing in Coxiella-infected cells.

CvpB mediates cathepsinB loss (Figure 3):

There is still substantial CTSB signal in *cvpB*-expressing cells (Figure 3A). Further, *cvpB* mutant-infected cells have intermediate CTSB levels by immunoblot (Figure 3C and 3F). When the CTSB signal is quantitated by microscopy to examine on a cellular level, is there a measurable change in cells expressing ectopic CvpB (Figure 3A) or cells infected with the *cvpB* mutant (not performed)? The intermediate phenotypes observed suggest that other mechanisms besides *cvpB* may be involved.

The logic behind focusing on *cvpB* is not convincing (lines 191-198), given that there are multiple Coxiella effectors which localize to endosomes, lysosomes, and other vesicular compartments where *cvpB* would be found.

Impact of increased lysosomal protein secretion on Coxiella infection (Figure 4 and 5):

This is an exciting finding and appears to occur in both epithelial cells and macrophage-like cells. While BrefeldinA shows that secretion of lysosomal proteins occurs through the secretory pathway, it is critical to established whether this process is essential for Coxiella infection.

Minor comments

Line 38- "retention of cathepsin B leads to defects..." – this statement is misleading, as the data is from overexpression of CTSB and does not show localization. "Overexpression" is more accurate.

Introduction - a description of cathepsin processing would be beneficial for the reader

Lines 84-89. The word "marginally" downplays that the pH shift from 4.8 to 5.2 is quite significant when considering 1) pH is a log scale; 2) activity of lysosomal proteases can be very different at 4.8 versus 5.2; 3) late endosomes have a pH ~5.2, thus suggesting the CCV pH is more similar to a late endosome; and 4) Coxiella is degraded at pH 4.8, so the shift from 5.2 to 4.8 is clearly detrimental.

Line 383-385. This has already been demonstrated, where Coxiella has been shown to alter degradative capacity of the cells by decreasing the number of acidic lysosomes in host cells and increasing a larger pool of CD63-positive endosomes with an average pH significantly more alkaline than lysosomes (PMID: 28293541 and PMID: 31869379).

Reviewer #2

(Remarks to the Author)

The manuscript entitled 'Coxiella burnetii manipulates lysosomal proteases to facilitate intracellular success' by Bird et al., explores how infection with the intracellular pathogen *C. burnetii* modifies abundance and trafficking events of lysosomal proteases, in particular Cathepsin B. The authors observe that Cathepsin B is actively removed from infected cells in a T4SS dependent manner, which was attributed to the secreted T4SS effector CvpB (Cig2). The functional relevance of Cathepsin B removal was demonstrated by the over-expression of Cathepsin B resulting in a reduction of intracellular *C. burnetii* bacterial loads. Finally, it was observed that immature Cathepsin B & D were detected extracellularly upon *C. burnetii* infection, with the former occurring independently of CvpB but instead require the default host cell secretory pathway.

The findings of this work are interesting and novel, particularly with respect to the *cvpB*-dependent regulation of intracellular cathepsin B levels. The advance in understanding the biological relevance of cathepsin forms trafficked extracellularly during *C. burnetii* infection independently of CvpB remains limited. Further examination of the events leading to cathepsin depletion and the functional role of cathepsin B activity, and exploration of the requirement for dot/ICM (and possible effectors) in extracellular cathepsin export, would strengthen the claims made by the authors.

Major points:

1. The authors should establish the subcellular localisation of CvpB during infection. To which organelle does CvpB localise? If CvpB is a secreted T4SS effector protein, how does it reach CtsB in the lysosomal compartment? The authors should address this in their model
2. The authors should establish the role of host cell death in the release of extracellular cathepsin B as well as the requirement for the *C. burnetii* T4SS.
3. The authors should test where CvpB alone is sufficient to reduce intracellular Cathepsin B levels. Does *cvpB* impact the activity of other cathepsins? E.g. as judged using the BMV109 activity-based probe? How does cathepsin B activity/abundance change in CvpB transfected cells?
4. If CvpB is capable of depleting CtsB levels, wouldn't one expect a lack of CvpB-CtsB co-localisation in Figure 3A? The

- authors observe the opposite. This should be addressed with respect to the working model
5. Does CvpB directly interact with CtsB or other proteins required for its trafficking? This could be tested by pulldown experiments.
 6. Can the authors exclude the role of the proteasome in CvpB-dependent CtsB depletion?
 7. Does BFA treatment affect *Coxiella* intracellular viability and/or proliferation (Fig 5c)?
 8. To what extent is cathepsin B (as compared to other cathepsins) required for *Coxiella* intracellular proliferation? Cathepsin over-expression does not rule out the importance of other cathepsins in regulating *Coxiella* intracellular proliferation or vacuolar mass. To compliment the CtsB over-expression data, the authors should test *Coxiella* proliferation in cells where Cathepsin B has been genetically perturbed or pharmacologically inhibited (e.g. CA-074Me)
 9. Is cathepsin B depletion the major function of CvpB? Is *Coxiella* vacuole biogenesis affected in *ctsB* *-/-* cells? Does pharmacological cathepsin B inhibition (e.g. CA-074Me) improve *Coxiella* intracellular proliferation? And finally, can vacuole biogenesis defects of a CvpB mutant be rescued by perturbing cathepsin B expression and/or activity?
 10. Using the same panel of transfected effectors as referred to in relation to Figure 3a, could the authors screen for effectors that regulate the extracellular CtsB export?
 11. At what level does the CtsB depletion occur, and does it appear to occur prior or subsequent to lysosomal fusion? Is it occurring at the post-transcriptional/translationally? Through direct cleavage, autophagy or Proteasomal degradation
 12. L311-315 – It could be tested whether NLRP3 activation is dampened in *Coxiella* infected cells in a CvpB-dependent manner? This could be measured by exogenously adding an NLRP3 activator and measuring IL-1B release in cell infected with wildtype and mutant *Coxiella*

Minor points:

13. Figure 1F – how do CTSB protein levels differ in a *icmL::Tn* and *icmS::Tn* mutant in infected cells?
14. Are their regions, domains, or putative active sites on CvpB that are required for the binding/degradation of Cathepsin B?
15. Figure 5c – upon infection with wildtype *Coxiella* (DMSO control) lysate, there appears to be an increase in CTSB(Pro-form). Is this CvpB dependent?
16. L192 – The authors should comment on what was the panel of effectors that were tested and what was the outcome for each.
17. L251-253 – The authors may consider examining cathepsin peptide level coverage along the length of the protein. Do those in the N-term versus C-terminus differ in their relative abundance, which could indicate potential differences in cellular processing? Could it be that PTMs (possibly pathogen-induced) are hinder the identification of some of the cathepsin B peptides?
18. L591 – Are the protein enrichment scores calculated using the observed proteome as a background?
19. Authors should comment on the apparent specificity of intracellular cathepsin depletion for Cathepsin B and why not other cathepsins.
20. Figure 1G – Indicate estimated MOIs on figure
21. Several Cathepsin B western blots have the HMW region cropped, it would be informative to not crop this section of the gel as it may provide additional information regarding Cathepsin B processing in the difference conditions. i.e. Fig 1c, 1g, 3c.
22. Figure 1F – Please define what the Asterix in the image refers to. If it refers to infected cells, please describe the definition of an infected cell in this context.
23. L38 – unclear what is meant by 'retention' in this context. Please clarify
24. L175 – As this is not a full restoration in the catalytic inactive mutant, I would suggest the phrasing is toned down, or alternatively comment on the degree to which the restoration was observed.
25. L234-237 – the authors did not comment on the absence of cathepsin B in these experiments. Was it not detectable in these experiments? Or does this relate to the fact that *Coxiella* downregulates cathepsin B?

Reviewer #3

(Remarks to the Author)

In the manuscript "*Coxiella burnetii* manipulates lysosomal proteases to facilitate intracellular success" by Lauren E. Bird et al., the authors showed that cells infected with *C. burnetii* lack cathepsin B. Overexpression of cathepsin B reduce the replication ability of *C. burnetii*, suggesting that the removal of cathepsin B is an important virulence mechanism. The removal of cathepsin B from the infected cells seems to be accomplished by at least two different mechanisms: i) general secretion of lysosomal proteins, which includes cathepsin B, into the extracellular milieu and ii) by a so far uncharacterized mechanism, which depends on the T4BSS effector protein CvpB. The authors could show that CvpB-mediated cathepsin B removal is not the result of CvpB-induced PIKfyve inhibition nor CvpB-activated secretion of lysosomal proteins.

The manuscript is well written and increases our knowledge about how *C. burnetii* is able to replicate inside a phagolysosome. This is a significant information in *C. burnetii* pathogenesis. However, this manuscript does not provide an idea about the mechanism how *C. burnetii* and/or CvpB mediates the removal of cathepsin B from the infected cells. The following points have to be address before a final decision can be made:

- Figure 1G: The authors tried to compensate for the defect of intracellular replication of the T4BSS mutant by increasing the MOI. However, judged by DotB levels, even with the highest MOI, they are far from reaching similar levels as wt. Thus, the lack of removal of cathepsin B from cells infected with the T4BSS mutant might be caused by a lower level of bacteria present. Thus, the conclusion that the T4BSS is involved in the process of cathepsin B removal can not be drawn from this experiment. Please try to adjust the bacterial load or discuss accordingly.
- Figure 1F: it would be important to see/stain the bacteria. Why are they not stained by DAPI?
- Figure 3C: it would be essential to include a control for bacterial loading. The CvpB mutant has a replication defect. Thus,

the observed difference in cathepsin B levels might be just caused by the different bacterial level.

- The data suggests that CvpB is involved in cathepsin B removal, but the authors do not provide any explanation how this might be accomplished. CvpB and cathepsin B might colocalize. Do they interact? How does CvpB mediate cathepsin B removal? It is not mediated by PIKfyve inhibition, nor by lysosomal protein secretion. Does CvpB expression influence the localization or stability of cathepsin B?

- The authors have shown that the general secretion of lysosomal proteins into the extracellular space is independent of CvpB. However, is this activity host cell or bacteria driven. Does a T4BSS mutant or dead bacteria also induce secretion of lysosomal proteins? What is the impact of the secreted lysosomal proteins on the neighboring cells? Are they more permissive for *C. burnetii* infections?

Version 1:

Reviewer comments:

Reviewer #1

(Remarks to the Author)

The authors have done an excellent job addressing comments from the first review.

A new finding in the revised manuscript is that *cvpB*-mediated reductions in cathepsin B levels result from a multi-vacuolar phenotype rather than a direct effect of *cvpB*. This suggests a role for CCV homotypic fusion, defects in which have been observed in numerous studies targeting either bacterial or host factors. Testing this model would be straightforward and could significantly enhance the study's impact, given that the role of homotypic fusion remains poorly understood.

The second key finding is that lysosomal protein secretion is a host cell response and occurs independently of the *Coxiella* T4SS. Therefore, the speculation that the cell is compensating for a *Coxiella* T4SS-dependent increase in host lysosomal protein synthesis (lines 489–498) contradicts both the data and the authors' conclusions. These lines should be removed or reframed, especially in light of recent evidence showing that the *Coxiella* T4SS actively inhibits TFEB activation and TFEB-mediated gene transcription (PMID: 39057917). This recent study also affects lines 470–474.

The authors demonstrate that *Coxiella* specifically manipulates cathepsin B but not other proteases, as stated in the discussion (lines 460–463). The other observed changes in lysosomal proteases (e.g., secretion) are a host cell response. Therefore, the title and abstract (lines 42–45) overstate the claim that *Coxiella* actively manipulates lysosomal proteases. This needs to be revised to reflect that only one protease, cathepsin B, is directly influenced by *Coxiella*.

Reviewer #2

(Remarks to the Author)

The authors have made a commendable effort addressing this reviewers' concerns with additional experiments, analysis and revising the text where appropriate.

This Reviewer has no more concerns.

Reviewer #3

(Remarks to the Author)

The majority of my concerns have been addressed in this revised version.

We wish to thank the reviewers for their insights, suggestions, and helpful feedback. We believe this has aided in improving the quality of the manuscript and the clarity of our research outcomes. A point by point response to the reviewer comments is included below.

REVIEWER COMMENTS

Reviewer #1 (Remarks to the Author):

In Bird et al, the authors examine manipulation of host lysosomal proteases by the obligate intracellular bacterial pathogen *Coxiella burnetii*. *Coxiella* survives in a modified phagolysosome (aka CCV) known to contain lysosomal hydrolases and proteases, yet the bacteria avoids degradation. This study further examines the role of host proteases and how *Coxiella* may be manipulating them during infection. The authors convincingly show that intracellular cathepsinB protein levels and activity are decreased in *Coxiella*-infected cells. Overexpression of cathepsinB has a modest effect on *Coxiella* replication and CCV size. The *Coxiella* secreted effector protein CvpB is implicated in manipulating cathepsinB levels, albeit through an unidentified mechanism. Finally, the authors discovered that secretion of lysosomal proteases is increased in *Coxiella*-infected cells, in a *cvpB*-independent manner. The resulting model proposes that *Coxiella* infection leads to increased secretion of lysosomal proteins to the extracellular space, while *cvpB* is involved in selectively removing cathepsinB from the CCV. Overall, the paper is well written, figures clearly presented and the experiments well-executed. The finding that *Coxiella* increases secretion of lysosomal proteins provides important insight into *Coxiella* infection. However, significant gaps remain, including mechanism for *cvpB* and evidence that the proposed model is critical for successful *Coxiella* infection.

Decreased cathepsinB in *Coxiella*-infected cells (Figure1):

The data is convincing that, in *Coxiella*-infected cells compared to uninfected cells, pro-CathepsinB is secreted to the extracellular space while there is a dramatic decrease in catalytically-active cathepsinB. While this is interpreted as cathepsinB is removed by the bacteria (specifically *cvpB*), it is also plausible *Coxiella* blocks general cathepsin processing through an indirect mechanism. Supporting this, in Figure 1C and 3A there appears to be an increase in pro-cathepsinD in infected cells suggesting a defect in cathepsinD processing; it's less clear whether cathepsinB processing is being altered as pro-cathepsinB is not shown, although in Figure 5C there does appear to be pro-cathepsinB in infected by not uninfected DMSO-treated cells. Others studies have demonstrated that *Coxiella*-infected cells have few lysosomes and an expanded pool of less acidic endosomes (PMID: 28293541 and PMID: 31869379), which could lead to a defect in cathepsin processing. The authors should test whether decreased levels of proteolytically active cathepsinB is due to the altered endosomal compartment in *Coxiella*-infected cells; one possible experiment is overexpression of TFEB to increase lysosomal biogenesis.

Response: We appreciate the reviewer's analysis of our work and thank them for raising these queries. All immunoblots have only been cropped in regions where no bands were present, but we have made a considered effort to ensure that all

cathepsin B blots now include the pro-region, even if there are no evident bands. Uncropped blots can also be found in source data.

The reviewer is correct in that there is a marked defect in cathepsin D processing during infection. As highlighted line 132-134 this is a result of cathepsin B being required for activation of cathepsin D.

We appreciate the suggestion that decreased cathepsin B could be due to an altered endosomal compartment and have now included work in Supplementary Fig 1G-H which demonstrates that overexpression of TFEB has no impact on cathepsin B loss during infection.

It would also be helpful to clarify whether both pro- and processed forms of cathepsinB are being detected in the mass spec, which portions the antibodies recognize for IFA and immunoblotting, and where the qRT-PCR primers are targeted.

We thank the reviewer for raising this point and have now clarified in the main text that our monoclonal cathepsin B antibody was raised against the heavy chain, thus it can recognize pro-CTSB, single chain and heavy chain forms (see line 130 in the revised text). We have also added a note with respect to the qRT-PCR primers in Table 1.

We agree that having information about both the pro- and mature cathepsin B abundance in the mass spectrometry data is helpful and include new data in which we sought to improve peptide coverage of cathepsin B protein with data-independent acquisition mass spec (DIA-MS) (Fig 1E). Using this approach, we were able to identify a peptide in the pro-domain of cathepsin B which showed the opposite trend to peptides from the mature protein. This verifies that cathepsin B is still being synthesised and is removed from cells after its processing to the mature form.

The representative image in Figure 1F is dramatic and impressive. Does the antibody used recognize only processed cathepsinB? Is there any cathepsinB in the cell? This data would be strengthened by quantitating cathepsinB signal in uninfected and infected cells for WT as well as *icmL* and *icmS* mutants. This is true of other figures, and both microscopy and immunoblotting should be quantitated whenever possible.

We thank the reviewer for their kind appraisal of our microscopy data (Fig 1G in the revised manuscript). As discussed above, the antibody used in this imaging recognises the heavy chain subunit of cathepsin B. We appreciate the reviewer's point that this data would be strengthened with quantitation and have included this data in Fig 1H.

With respect to staining for cathepsin B during infection with *icmL::Tn* and *icmS::Tn* mutants, this was attempted but limited by low bacterial load making it difficult to detect infected cells (new Supplementary Fig 2). To compensate for this, we repeated and quantified our western blot approach (Fig 1I,1J) in an attempt to ensure equivalent levels of intracellular bacteria between WT, *icmL::Tn* and *icmS::Tn* at the time of sample collection.

We also acknowledge the reviewer's point that microscopy and western blotting would benefit from quantitation and have included this in our manuscript where feasible.

Overexpression of CTSB is detrimental to *Coxiella* (Figure 2):

Figure 2D and line 175: The catalytically-inactive CTSB leads to an intermediate phenotype, as there is no significant difference between the catalytically inactive CTSB and either GFP or active CTSB. Thus, the statement that *Coxiella* "replication was lower in CTSB-GFP cells than cells expressing GFP alone....this was restored when cathepsinB was catalytically inactive" is overinterpreted. This intermediate phenotype is also seen in CCV size. The intermediate phenotype of the catalytically inactive CTSB, especially at 5 dpi, suggests that there might be a time-dependent role for CTSB during infection. Measuring viable bacteria using colony forming unit (CFU) assays instead of genome equivalents would generate more accurate insight into the effect of cathepsinB on *Coxiella*.

We thank the reviewer for their analysis of these data. We have revised our manuscript with a conscious effort not to overstate this phenotype but instead emphasise the intermediate phenotype observed with respect to *Coxiella* replication in CTSB^{C108A} expressing cells.

We acknowledge the reviewer's suggestion to supplement this data with CFU assays however have not found these assays to yield reliable/reproducible results in our laboratory, at least in part due to the small colony size of *C. burnetii*. Use of qPCR to determine genome equivalents is a well-established method for measuring *Coxiella* intracellular replication (PMIDs 15489446, 20515926, 25422265, 12734219 amongst others) and we believe this method is appropriate for our purposes. If anything, we suspect that our approach here may actually underestimate the detrimental impact of overexpressing CTSB as we are capturing both live and dead bacteria through this method.

Figure 2: Please clarify SC and HC in the figure legend (2B) and indicate where the processing events occur (2A).

We thank the reviewer for this suggestion and have indicated the subcellular compartment in which processing occurs in Fig 2A. SC represents the 'single chain' of cathepsin B, while HC refers to 'heavy chain' – we have clarified this in our new figure legend for Fig 2.

2B and 2C: Have GFP immunoblots been performed, instead of total CTSB? This might provide insight into CTSB processing in *Coxiella*-infected cells.

We performed immunoblots to GFP now included in Fig 2C. We were intrigued to see a band at the same size as free GFP in infected cells, suggesting the CTSB-GFP fusion protein is being cleaved in an infection-dependent manner. However, the abundance of the fusion protein (approx. 75 kDa) does not decrease accordingly, nor do we observe a noticeable size shift that would correlate with infection-dependent processing. We have added this observation to the manuscript text.

CvpB mediates cathepsinB loss (Figure 3):

There is still substantial CTSB signal in *cvpB*-expressing cells (Figure 3A). Further, *cvpB* mutant-infected cells have intermediate CTSB levels by immunoblot (Figure 3C and 3F). When the CTSB signal is quantitated by microscopy to examine on a cellular level, is there a measurable change in cells expressing ectopic CvpB (Figure 3A) or cells infected with the *cvpB* mutant (not performed)? The intermediate phenotypes observed suggest that other mechanisms besides *cvpB* may be involved.

We thank the reviewer for raising these points and agree that there are other mechanisms involved in CvpB-mediated CTSB depletion. Figure 3 has now been significantly reworked based on new data.

Quantitation of CTSB during infection with the *cvpB* mutant has now been performed and can be found in Fig 3C and D, where we observed a similar trend to our western blot data (ie that some cathepsin B is retained during infection with *cvpB::Tn*, but less than what we find in uninfected cells). Additionally, we have quantified CTSB signal in cells ectopically expressing CvpB (Fig 3K) and found that these were not significantly different from control, suggesting that the reviewer is correct that this relationship is not direct and instead represents an indirect outcome of CvpB function during infection. We thank the reviewer for strengthening our manuscript with this suggestion.

The logic behind focusing on *cvpB* is not convincing (lines 191-198), given that there are multiple *Coxiella* effectors which localize to endosomes, lysosomes, and other vesicular compartments where *cvpB* would be found.

We acknowledge the reviewer's point and have revised the order in which our data is presented. We initially screened our library of *Coxiella* effector mutants, however this library is not comprehensive and only allowed us to assess the impact of 38/~150 *Coxiella* effectors. We tried to get around this by individually assessing the contribution of individual effectors through ectopic expression, where we could investigate the impact of a larger library of effectors. We agree that the co-localisation between CvpB and CTSB at endosomal or vesicular compartments is perhaps unsurprising, and – as indicated above – our data demonstrates that CvpB does not directly degrade CTSB but loss of CTSB is linked to the infection/CCV environment which is altered in the *cvpB* mutant.

Impact of increased lysosomal protein secretion on *Coxiella* infection (Figure 4 and 5):

This is an exciting finding and appears to occur in both epithelial cells and macrophage-like cells. While BrefeldinA shows that secretion of lysosomal proteins occurs through the secretory pathway, it is critical to established whether this process is essential for *Coxiella* infection.

We thank the reviewer for their positivity regarding this finding. We now include data in our revised Figure 5 which aims to assess whether the secretion of lysosomal content is important for *Coxiella* infection and discuss it below.

We quantified intracellular bacterial load after brefeldin A treatment and saw no difference in the amount of Coxiella present (Fig 5F), suggesting that inhibition of this pathway is not immediately detrimental to Coxiella. This aligns with a previous report that GNPTAB^{-/-} cells show no difference in replication (PMID 31405956). Although not directly assessed in that study, more recent work suggests that GNPT depletion will cause redirection of hydrolases to the extracellular space (PMIDs 36074821, 36074822). The observation that replication is unchanged in GNPT cells therefore suggests that hydrolase trafficking and/or secretion is dispensable for Coxiella replication.

To further assess this in our hands, we designed an experiment in which naïve cells were treated with the secretome from mock-infected or *Coxiella*-infected cells. We saw no differences in Coxiella replication, suggesting that the secretome is not changing susceptibility of neighbouring cells to infection. However, the dynamics of this experiment are complex and we cannot rule out that secreted lysosomal proteins impact the localised host response to infection.

We agree with the reviewer that this is an exciting finding, and further investigation into this phenotype is ongoing in our laboratory.

Minor comments

Line 38- “retention of cathepsin B leads to defects...” – this statement is misleading, as the data is from overexpression of CTSB and does not show localization. “Overexpression” is more accurate.

We thank the reviewer for highlighting this and have rewritten accordingly.

Introduction - a description of cathepsin processing would be beneficial for the reader

We thank the reviewer for this suggestion to strengthen our manuscript and have added this into the introduction (see lines 55-60 in the updated manuscript)

Lines 84-89. The word “marginally” downplays that the pH shift from 4.8 to 5.2 is quite significant when considering 1) pH is a log scale; 2) activity of lysosomal proteases can be very different at 4.8 versus 5.2; 3) late endosomes have a pH ~5.2, thus suggesting the CCV pH is more similar to a late endosome; and 4) Coxiella is degraded at pH 4.8, so the shift from 5.2 to 4.8 is clearly detrimental.

We agree with the reviewer that this language understates the shift in lysosomal environment and have removed the word ‘marginally’.

Line 383-385. This has already been demonstrated, where Coxiella has been shown to alter degradative capacity of the cells by decreasing the number of acidic lysosomes in host cells and increasing a larger pool of CD63-positive endosomes with an average pH significantly more alkaline than lysosomes (PMID: 28293541 and PMID: 31869379).

We appreciate the reviewer for raising this important point and agree. We have added these citations to our manuscript accordingly.

Reviewer #2 (Remarks to the Author):

The manuscript entitled 'Coxiella burnetii manipulates lysosomal proteases to facilitate intracellular success' by Bird et al., explores how infection with the intracellular pathogen *C. burnetii* modifies abundance and trafficking events of lysosomal proteases, in particular Cathepsin B. The authors observe that Cathepsin B is actively removed from infected cells in a T4SS dependent manner, which was attributed to the secreted T4SS effector CvpB (Cig2). The functional relevance of Cathepsin B removal was demonstrated by the over-expression of Cathepsin B resulting in a reduction of intracellular *C. burnetii* bacterial loads. Finally, it was observed that immature Cathepsin B & D were detected extracellularly upon *C. burnetii* infection, with the former occurring independently of CvpB but instead require the default host cell secretory pathway.

The findings of this work are interesting and novel, particularly with respect to the cvpB-dependent regulation of intracellular cathepsin B levels. The advance in understanding the biological relevance of cathepsin forms trafficked extracellularly during *C. burnetii* infection independently of CvpB remains limited. Further examination of the events leading to cathepsin depletion and the functional role of cathepsin B activity, and exploration of the requirement for dot/ICM (and possible effectors) in extracellular cathepsin export, would strengthen the claims made by the authors.

We thank the reviewer for their thorough appraisal of our work and address their suggestions as outlined below.

Major points:

1. The authors should establish the subcellular localisation of CvpB during infection. To which organelle does CvpB localise? If CvpB is a secreted T4SS effector protein, how does it reach CtsB in the lysosomal compartment? The authors should address this in their model

We thank the reviewer for raising this important point. CvpB has previously been demonstrated to localise to the CCV during infection (PMIDs 25422265, 27226300) and we have now added this to text.

We have also significantly reworked our figure on the interaction between CvpB and cathepsin B (see revised Fig 3). We did not see a direct physical interaction between CTSB and CvpB with a pulldown experiment (Supplementary Fig 3E) and believe that retention of CTSB during infection with the cvpB mutant is a result of an altered CCV compartment compared to the CCV formed during WT infection.

2. The authors should establish the role of host cell death in the release of extracellular cathepsin B as well as the requirement for the *C. burnetii* T4SS.

We thank the reviewer for their suggestion. As outlined in Fig 4A, we saw equivalent amounts of actin present in the media from uninfected and infected cells, suggesting

no major difference in the amount of cell lysis and we therefore do not believe host cell death is responsible for the increased secretion of cathepsin B we see during infection. *Coxiella* is also known to translocate multiple effectors that inhibit both apoptotic and pyroptotic cell death pathways as means of maintaining its replicative niche (PMID 32958854, 23126667, 26687278), making it unlikely that cathepsin secretion is a result of increased host cell death during infection.

We also appreciate the reviewer's second point that the requirement of the T4SS for lysosomal protein secretion should be investigated. We now include data in Figure 5 (Fig 5D-E) which indicate that secretion is increased from uninfected cells even during infection with a T4SS mutant (*icmL::Tn*), suggesting this process is not driven by the T4SS. This is an important new finding as it indicates that lysosomal protein secretion may be a host directed response to infection rather than a consequence of pathogen virulence. We have updated our manuscript accordingly.

3. The authors should test where CvpB alone is sufficient to reduce intracellular Cathepsin B levels. Does *cvpB* impact the activity of other cathepsins? E.g. as judged using the BMV109 activity-based probe? How does cathepsin B activity/abundance change in CvpB transfected cells?

We thank the reviewer for raising this important point. As discussed above, we have now included data in Fig 3 which demonstrate that ectopic expression of CvpB is not sufficient to alter cathepsin B levels (Fig 3K), which strengthens our hypothesis that the interaction between these two proteins is indirect.

We also now include data in the manuscript to demonstrate that infection with the CvpB mutant leads to specific changes in the abundance and activity of cathepsin B, but not other cysteine cathepsins as assessed by BMV109 labelling (Fig 3F).

4. If CvpB is capable of depleting CtsB levels, wouldn't one expect a lack of CvpB-CtsB co-localisation in Figure 3A? The authors observe the opposite. This should be addressed with respect to the working model

As discussed above, the new data we have added to the manuscript now indicate that CTSB retention is a property of infection with *cvpB::Tn* and not a direct effect of CvpB itself.

5. Does CvpB directly interact with CtsB or other proteins required for its trafficking? This could be tested by pulldown experiments.

We thank the reviewer for their helpful suggestion. Multiple independent pulldown approaches were attempted during revision of this manuscript, and we now include a new figure in Supplementary Fig 3E which shows no direct interaction between CTSB and CvpB. We utilised the same approach for IP-MS experiments and successfully pulled down other CvpB interactors which are currently under further investigation in our laboratory and we don't believe they are required to support the conclusions drawn from this manuscript with respect to cathepsin B.

6. Can the authors exclude the role of the proteasome in CvpB-dependent CtsB depletion?

We found that MG132 treatment had no effect on cathepsin B dynamics in either wild-type or *cvpB::Tn* infection. This result is now included in Supplementary Fig 3H-I.

7. Does BFA treatment affect Coxiella intracellular viability and/or proliferation (Fig 5c)?

We thank the reviewer for raising this point. Because BFA is toxic when applied to mammalian cells, we applied only a short time course (6hrs) for our BFA treatments. Per the reviewer's comment, we assessed Coxiella intracellular load following 6 hrs of treatment and found no significant difference between BFA treatment and the control.

8. To what extent is cathepsin B (as compared to other cathepsins) required for Coxiella intracellular proliferation? Cathepsin over-expression does not rule out the importance of other cathepsins in regulating Coxiella intracellular proliferation or vacuolar mass. To compliment the CtsB over-expression data, the authors should test Coxiella proliferation in cells where Cathepsin B has been genetically perturbed or pharmacologically inhibited (e.g. CA-074Me)

We thank the reviewer for raising these points and agree that modulation of one cathepsin can have myriad impacts on the abundance and activity of other cathepsins (PMIDs 26773004, 38410910)

Per the reviewer's request, we applied the selective cathepsin B inhibitor Ca074Me to cells and measured proliferation over a timecourse (Fig 3G and H). We did not observe any change to Coxiella replication in the presence of this inhibitor, which we believe is likely because cathepsin B is already undetectable in cells at 1-2 days following infection, so there is likely no protein remaining to inhibit.

9. Is cathepsin B depletion the major function of CvpB? Is Coxiella vacuole biogenesis affected in *ctsB* *-/-* cells? Does pharmacological cathepsin B inhibition (e.g. CA-074Me) improve Coxiella intracellular proliferation? And finally, can vacuole biogenesis defects of a CvpB mutant be rescued by perturbing cathepsin B expression and/or activity?

CvpB has been partially characterised as having a major role in vacuole biogenesis (PMID 27226300). As stated above, we believe that cathepsin B depletion is not the major function of CvpB but more likely an outcome of the altered CCV environment produced in the absence of CvpB.

As discussed above, we saw no change to Coxiella replication (of either WT or *cvpB::Tn*) in the presence of Ca074Me providing further evidence of this indirect relationship. CvpB has a core role in establishing the ideal replication environment for Coxiella.

10. Using the same panel of transfected effectors as referred to in relation to Figure 3a, could the authors screen for effectors that regulate the extracellular CtsB export?

We thank the reviewer for this suggestion. However, as mentioned above, we have now added data to the manuscript where we observe increased secretion even during infection with *icmL::Tn*, suggesting that this phenotype is not mediated by Dot/Icm effector activity.

11. At what level does the CtsB depletion occur, and does it appear to occur prior or subsequent to lysosomal fusion? Is it occurring at the post-transcriptional/translationally? Through direct cleavage, autophagy or Proteasomal degradation

We thank the reviewer for raising these questions. We believe cathepsin B is removed from cells post-transcriptionally, given that we do not see a decrease in transcript abundance in our RT-qPCR data (Fig 1D), and new DIA-MS data with improved peptide coverage demonstrates that a peptide from the pro-form is not decreased during infection (Fig 1E). Given that only the mature protein shows decreased abundance during infection, this would indicate that the protein is being processed in the lysosome and then the mature protein is removed.

With respect to the reviewer's other comments, we showed that MG132 treatment had no impact on cathepsin B dynamics, indicating CTSB loss is not via the proteasome (Supplementary Fig 3H-I), and identified no direct interaction between CTSB and CvpB, demonstrating that CvpB is not directly cleaving CTSB. We speculate that CTSB removal is due to the vacuolar environment established by CvpB, though the exact mechanism is still under investigation in our laboratory.

12. L311-315 – It could be tested whether NLRP3 activation is dampened in Coxiella infected cells in a CvpB-dependent manner? This could be measured by exogenously adding an NLRP3 activator and measuring IL-1B release in cell infected with wildtype and mutant Coxiella

We appreciate the reviewer's suggestion. To our knowledge, many NLRP3 agonists rely on cathepsin activity upstream of inflammasome activation and we feel that the results from the experiment suggested above would be difficult to interpret given our data which demonstrates differences in cathepsin abundance between uninfected and infected cells. While an interesting area for investigation, we do not feel that addressing this question is central to the current manuscript.

Minor points:

13. Figure 1F – how do CTSB protein levels differ in a *icmL::Tn* and *icmS::Tn* mutant in infected cells?

We thank the reviewer for raising this. In our updated manuscript, we include a new Supplementary Figure (Supplementary Fig 2) depicting microscopy during infection with *icmL::Tn* and *icmS::Tn*. As evident in this imaging, it was difficult to determine which cells were infected on the basis of LAMP1 staining due to the lack of intracellular replication in these strains. We therefore believe that western blotting and quantification of CTSB at the whole population level to be a more appropriate approach to addressing the reviewer's point.

In our updated manuscript, we have replaced the previous figure 1F with a new blot (1I) and quantification (1J). We spent time optimising our infection durations and MOIs in order to compensate for the differences in intracellular replication between these strains. We decreased the baseline MOI which we used for WT infection, and shortened the duration of infection to 2 days. Using this approach, we were able to get equivalent amounts of intracellular bacteria in WT, *icmL::Tn* and *icmS::Tn* mutants, as indicated by our anti-Coxiella western blots. With this newly optimised infection protocol, we observed that cathepsin B is retained during infection with the *icmL::Tn* strain compared to WT, even with increasing MOI. However, it is lost during infection with *icmS::Tn*. Collectively, this indicates that CTSB loss is linked to Dot/Icm activity but is independent of the *Coxiella* IcmS chaperone.

14. Are their regions, domains, or putative active sites on CvpB that are required for the binding/degradation of Cathepsin B?

We thank the reviewer for this question. To date, the structure of CvpB – or any *Coxiella* effector – has not been solved and low confidence AlphaFold models limit the utility of prediction algorithms. We have emphasised this point in the main text of our manuscript (line 363-364).

However, we agree with the reviewer that more information about regions of CvpB that are important for CTSB removal is a useful addition to our manuscript. We now include Supplementary Fig 4, where we infected cells with *Coxiella* strains which had a transposon in CvpB but had been complemented with various N-terminal truncations. We did not truncate CvpB from the C-terminus as this region of the protein is required for the protein to be translocated through the T4SS.

Interestingly, cathepsin B was retained during infection with the truncated complementations, but not the full length (Supplementary Fig 4B,C), suggesting that the N-terminus of CvpB is required for cathepsin B removal. However, we also observed that none of these truncated complementations restored the multi-vacuole phenotype observed with CvpB disruption, suggesting again that cathepsin B removal is linked to differences in the CCV environment observed during infection with WT *Coxiella* or the CvpB mutant, respectively.

15. Figure 5c – upon infection with wildtype *Coxiella* (DMSO control) lysate, there appears to be an increase in CTSB(Pro-form). Is this CvpB dependent?

We thank the reviewer for this observation. Depending on the exposure conditions in our immunoblots, we sometimes observed a faint band corresponding to pCTSB during infection with WT *Coxiella*, consistent with our observation in Figure 1 that the pro-form is not decreased during infection (discussed above).

We draw the reviewer's attention to the immunoblots included in Supplementary Fig 3 (3C, 3F, 3H), where a faint band is observed at ~37kDa. This doesn't appear to be altered in the *cvpB::Tn* mutant, suggesting it is not CvpB dependent.

16. L192 – The authors should comment on what was the panel of effectors that were tested and what was the outcome for each.

We thank the reviewer for this suggestion and have now included an additional Supplementary Table 3 which lists which effectors were tested, what method they were tested by (western blotting or immunofluorescence microscopy) and whether we observed any change to cathepsin B expression.

17. L251-253 – The authors may consider examining cathepsin peptide level coverage along the length of the protein. Do those in the N-term versus C-terminus differ in their relative abundance, which could indicate potential differences in cellular processing? Could it be that PTMs (possibly pathogen-induced) are hindering the identification of some of the cathepsin B peptides?

We thank the reviewer for this suggestion. In Figure 1, we have now included a heatmap of peptide abundance across cathepsin B (Fig 1E) obtained from DIA-MS. As suggested by the reviewer, we did observe a difference in relative abundance of cathepsin B from the pro-domain at the N-terminus of the protein compared to peptides within the mature protein.

If certain PTMs were hindering identification of cathepsin B peptides, we would expect to see some unmodified peptides with no change to their relative expression. However, as demonstrated in Fig 1E, the decrease in abundance is observed for every peptide in the mature protein, making it unlikely that we are misidentifying peptides on the basis of modifications.

18. L591 – Are the protein enrichment scores calculated using the observed proteome as a background?

We thank the reviewer for raising this point. Enrichment scores were calculated using the *H. sapiens* protein coding genome as a background and this has been clarified in the methods.

19. Authors should comment on the apparent specificity of intracellular cathepsin depletion for Cathepsin B and why not other cathepsins.

We thank the reviewer for raising this and direct them to lines 588-602 in the discussion as follows:

“Why *C. burnetii* infection promotes the loss of cathepsin B over other cathepsins remains unclear. We did not observe a decrease in other proteases of the cathepsin family except for a slight reduction, but not complete absence, in cathepsin C expression at 3 days post infection (Fig 1C). It is important to note that a comprehensive profile of cathepsin B substrates remains elusive, however proteomic methods to identify carboxypeptidase substrates continue to improve and will likely provide an opportunity to elucidate the specific cathepsin B substrates that impact CCV biogenesis. This may aid in elucidating why cathepsin B is actively and selectively removed from *C. burnetii*-infected cells in contrast to other cathepsins. Interestingly, Qi et al. showed that genetic deletion of cathepsin B, but not cathepsin G or other non-lysosomal proteases, conferred enhanced resistance to the cytosolic bacterium *Francisella novicida*⁴⁹. The authors demonstrated that cathepsin B deletion promotes lysosomal biogenesis and autophagy by governing the activities of transcription factor EB (TFEB) and the

autophagy kinase ULK1. It is therefore plausible that the induction of TFEB activity^{21,50} and engagement of the autophagic pathway^{9,51} observed during *C. burnetii* infection could be a downstream effect of the cathepsin B loss we observe in our study”

20. Figure 1G – Indicate estimated MOIs on figure

Amended, with thanks.

21. Several Cathepsin B western blots have the HMW region cropped, it would be informative to not crop this section of the gel as it may provide additional information regarding Cathepsin B processing in the difference conditions. i.e. Fig 1c, 1g, 3c.

We appreciate the reviewer raising this point and agree. All cathepsin B blots now include the pro-region, even if no visible bands are present.

22. Figure 1F – Please define what the Asterix in the image refers to. If it refers to infected cells, please describe the definition of an infected cell in this context.

We thank the reviewer for raising this important point and have elaborated on this in the figure legend for Fig 1. The asterisk indicates an infected cell which was identified on the basis of LAMP-1 staining around the CCV periphery.

23. L38 – unclear what is meant by ‘retention’ in this context. Please clarify

We thank the reviewer for this observation and have changed our wording to ‘overexpression’ which we feel is more accurate.

24. L175 – As this is not a full restoration in the catalytic inactive mutant, I would suggest the phrasing is toned down, or alternatively comment on the degree to which the restoration was observed.

Rewritten, with thanks.

25. L234-237 – the authors did not comment on the absence of cathepsin B in these experiments. Was it not detectable in these experiments? Or does this relate to the fact that *Coxiella* downregulates cathepsin B?

We thank the reviewer for highlighting this important point. At the protein level, cathepsin B was not identified to be significantly altered in our mass spec experiments on the supernatant, in contrast to our western blots which clearly show an increased extracellular abundance of pro-cathepsin B. Because of this, we investigated the individual cathepsin B peptides identified in our mass spec data. We observed that the only significantly altered peptide is in the pro-domain of CTSB, aligning with our immunoblot data. We have now included this peptide heatmap as Fig 4C.

Reviewer #3 (Remarks to the Author):

In the manuscript "*Coxiella burnetii* manipulates lysosomal proteases to facilitate

intracellular success" by Lauren E. Bird et al., the authors showed that cells infected with *C. burnetii* lack cathepsin B. Overexpression of cathepsin B reduce the replication ability of *C. burnetii*, suggesting that the removal of cathepsin B is an important virulence mechanism. The removal of cathepsin B from the infected cells seems to be accomplished by at least two different mechanisms: i) general secretion of lysosomal proteins, which includes cathepsin B, into the extracellular milieu and ii) by a so far uncharacterized mechanism, which depends on the T4BSS effector protein CvpB. The authors could show that CvpB-mediated cathepsin B removal is not the result of CvpB-induced PIKfyve inhibition nor CvpB-activated secretion of lysosomal proteins.

The manuscript is well written and increases our knowledge about how *C. burnetii* is able to replicate inside a phagolysosome. This is a significant information in *C. burnetii* pathogenesis. However, this manuscript does not provide an idea about the mechanism how *C. burnetii* and/or CvpB mediates the removal of cathepsin B from the infected cells.

We thank the reviewer for their thorough appraisal of our work. We have addressed their concerns as outlined below:

The following points have to be address before a final decision can be made:

- Figure 1G: The authors tried to compensate for the defect of intracellular replication of the T4BSS mutant by increasing the MOI. However, judged by DotB levels, even with the highest MOI, they are far from reaching similar levels as wt. Thus, the lack of removal of cathepsin B from cells infected with the T4BSS mutant might be caused by a lower level of bacteria present. Thus, the conclusion that the T4BSS is involved in the process of cathepsin B removal can not be drawn from this experiment. Please try to adjust the bacterial load or discuss accordingly.

We agree with the reviewer and have modified this experiment to obtain similar bacterial loads between WT, *icmL::Tn*, and *icmS::Tn* strains. We did this by lowering the baseline MOI for WT to 1 and harvesting cells at 2dpi. As evident in Fig 1I anti-*Coxiella* blot, there is an equivalent amount of bacteria present between WT (MOI 1) and *icmL::Tn* (MOI 100), however there is still a statistically significant difference in CTSB density (quantified in Fig 1J).

- Figure 1F: it would be important to see/stain the bacteria. Why are they not stained by DAPI?

We thank the reviewer for raising this point. The bacteria are indeed stained by DAPI which can be observed when the image is overexposed (see the below DAPI image from Fig 1G). This is also apparent in our manuscript when screen brightness is at maximum (Fig 1G). While this is not ideal for imaging we were able to use this DAPI staining to facilitate quantification of this microscopy (now presented in Fig 1H).

- Figure 3C: it would be essential to include a control for bacterial loading. The CvpB mutant has a replication defect. Thus, the observed difference in cathepsin B levels might be just caused by the different bacterial level.

We thoroughly appreciate the reviewer raising this important point. To date, the literature is not consistent with regards to whether the CvpB mutant has a replication defect. Some groups claim the bacteria replicate to equal numbers as wild type (PMIDs 25080348, 27226300), while others claim there is a decrease in the CvpB mutant (PMIDs 25422265, 39560384). We aimed to assess the contribution of cell type (macrophage vs epithelial cell) and starting MOI in our laboratory. As included in Supplementary Fig 3A&B, we observed delayed replication in the *cvpB* mutant which was capable of reaching equivalent levels as WT once replication had plateaued in THP-1s. In HeLa cells, replication was decreased at D5, but still appeared to be in the exponential phase.

Given that we did observe a difference in replication dynamics, we infected cells with *cvpB::Tn* at increasing MOIs to WT (Supplementary Fig 3C). With increasing MOI we did observe a decrease in CTSB, but not a complete removal. In light of this data (and new data included our revised manuscript) we believe that CvpB is required to establish a CCV environment that lends itself to cathepsin B loss – whether by the action of another effector or a host factor such as altered lysosomal pH or ion concentration. These hypotheses have now been made clear throughout our revised manuscript.

- The data suggests that CvpB is involved in cathepsin B removal, but the authors do not provide any explanation how this might be accomplished. CvpB and cathepsin B might colocalize. Do they interact? How does CvpB mediate cathepsin B removal? It is not mediated by PIKfyve inhibition, nor by lysosomal protein secretion. Does CvpB expression influence the localization or stability of cathepsin B?

We thank the reviewer for raising these questions. Our revised manuscript includes a substantially reworked version of Figure 3 which addresses some of the reviewer's above points.

Briefly, we now demonstrate that modulation of cathepsin B by CvpB is not due to a direct interaction or binding. Ectopic expression of CvpB was not sufficient to modulate cathepsin B levels (Fig3 I, J and K), and we did not identify a physical interaction with pulldown experiments (Supp Fig 3E). Rather, we now propose that retention of cathepsin B during infection with the *cvpB* mutant is a result of an altered CCV environment, though the exact mechanism leading to cathepsin B retention is not yet clear.

- The authors have shown that the general secretion of lysosomal proteins into the extracellular space is independent of CvpB. However, is this activity host cell or bacteria driven. Does a T4BSS mutant or dead bacteria also induce secretion of lysosomal proteins? What is the impact of the secreted lysosomal proteins on the neighboring cells? Are they more permissive for *C. burnetii* infections?

We appreciate these suggestions by the reviewer. In our updated manuscript, we now demonstrate that secretion occurs during infection with *icmL::Tn* (to similar levels as WT) and seems to increase with increasing MOI (Fig 5D,E). We therefore conclude that the secretion is likely an intriguing host response to infection as it does not require a functional T4SS and is insensitive to differences in intracellular replication.

We were also able to demonstrate that naïve cells do not appear to be more susceptible to infection after being treated with the secretome (Fig 5J), at least in a cell culture context under the temporal dynamics tested. Work is ongoing in our laboratory to investigate whether the secretome has an impact on the transcriptome of neighbouring cells or inflammation in a tissue context.

REVIEWER COMMENTS

Reviewer #1 (Remarks to the Author):

The authors have done an excellent job addressing comments from the first review.

A new finding in the revised manuscript is that *cvpB*-mediated reductions in cathepsin B levels result from a multi-vacuolar phenotype rather than a direct effect of *cvpB*. This suggests a role for CCV homotypic fusion, defects in which have been observed in numerous studies targeting either bacterial or host factors. Testing this model would be straightforward and could significantly enhance the study's impact, given that the role of homotypic fusion remains poorly understood.

We thank the reviewer for their positive feedback. We agree that examining whether the multivacuole phenotype is directly linked to cathepsin B is an interesting area for further investigation, and we have addressed this hypothesis in the discussion of our paper (lines 447-451):

A multi-vacuole phenotype has also been observed in the absence of autophagy-related proteins (e.g. ATG5, STX17)^{29,30,48}. However, it has never been demonstrated that the CCVs formed in the absence of autophagy have the same properties as CCVs formed in the absence of *CvpB*. Future studies should investigate this question and test whether retention of cathepsin B is directly linked to defects in homotypic fusion.

As now stated in the discussion, a multi-vacuole phenotype has been observed in the absence of autophagy-related proteins (PMIDs: 25080348, 23362322, 27226300). Similarly, multiple CCVs are more frequent with siRNA gene silencing of clathrin heavy chain (PMID: 29973118). All of these different host factors are assumed to aid in the fusion of autophagosomes with the CCV, however there has never been a clear analysis demonstrating that the CCVs formed in the absence of autophagy/autophagic-lysosome fusion are the same as CCVs formed in the absence of *CvpB*. Additionally, the relationship between cathepsin B and autophagy is complex with research also demonstrating that cathepsin B can negatively regulate autophagy via TFEB (PMID: 27551156). It is plausible that removal of cathepsin B, in a *CvpB*-dependent manner, helps drive autophagy during infection and not the other way around. It is also likely that silencing autophagy-related proteins will have myriad effects on cathepsin expression and activity independent of infection, which may further confound results.

Given these complexities, we politely decline to include this further experimental work as we are concerned that the results from an experiment testing the link between cathepsin B and autophagy-related proteins will be difficult to interpret without significant additional analysis.

We do agree that understanding the purpose of homotypic fusion in CCV biogenesis is important. Based on other data that was generated during the preparation of this manuscript, our laboratory is still actively investigating the mechanistic role of *CvpB*, and we will address the reviewer's hypothesis in our follow up research into this important bacterial effector.

The second key finding is that lysosomal protein secretion is a host cell response and occurs independently of the *Coxiella* T4SS. Therefore, the speculation that the cell is compensating for a *Coxiella* T4SS-dependent increase in host lysosomal protein synthesis (lines 489–498) contradicts both the data and the authors' conclusions. These lines should be removed or reframed, especially in light of recent evidence showing that the *Coxiella* T4SS actively inhibits TFEB activation and TFEB-mediated gene transcription (PMID: 39057917). This recent study also affects lines 470–474.

We thank the reviewer for highlighting this and have amended our text (line 478-484):

"While *C. burnetii* infection is known to promote engagement of the autophagic pathway^{9,51}, there is conflicting literature on whether TFEB is activated^{21,52} or inhibited⁵³ during infection in a T4SS-dependent manner. It is likely that the true nature of TFEB signalling during infection is dynamic and influenced by multiple factors (as reviewed in⁵). Nonetheless, it is plausible that the reported increase in lysosomal biogenesis and autophagy observed during *C. burnetii* infection could be a downstream effect of the cathepsin B loss we observe in our study."

Speculation about M6PR saturation as a result of TFEB activation has been removed in light of the conflicting evidence regarding TFEB signalling during infection.

The authors demonstrate that *Coxiella* specifically manipulates cathepsin B but not other proteases, as stated in the discussion (lines 460–463). The other observed changes in lysosomal proteases (e.g., secretion) are a host cell response. Therefore, the title and abstract (lines 42–45) overstate the claim that *Coxiella* actively manipulates

lysosomal proteases. This needs to be revised to reflect that only one protease, cathepsin B, is directly influenced by *Coxiella*.

We thank the reviewer for pointing this out and have amended our title and abstract to more accurately reflect that *Coxiella* specifically modulates cathepsin B but not other proteases.

Reviewer #2 (Remarks to the Author):

The authors have made a commendable effort addressing this reviewers' concerns with additional experiments, analysis and revising the text where appropriate.

This Reviewer has no more concerns.

We thank the reviewer for their helpful suggestions to strengthen our manuscript.

Reviewer #3 (Remarks to the Author):

The majority of my concerns have been addressed in this revised version.

We thank the reviewer for their helpful suggestions to strengthen our manuscript.